# Multi-omics and pan-cancer analysis revealed common molecular signatures to disclose multitargeted anticancer agents through network pharmacology approach

Hriddhi Sarker[1,2], Farhad Bin Farid[1,3], Marguba Kamrun[1,4], Esha Masud[1,5], Asif Ahmed[1,5], Mamun Miah[1,6], Neladre Shaker Roy[1], Neeraj Kumar[7], Md Ahad Ali[1,8]*

**1** Department of Computational Chemistry and Drug Design, Panacea Research Center, Rajshahi, Bangladesh, **2** Department of Biochemistry and Molecular Biology, University of Rajshahi, Rajshahi, Bangladesh, **3** Department of Pharmacy, University of Development Alternative (UODA), Dhaka, Bangladesh, **4** Department of Chemistry and Biochemistry, University of Oklahoma Norman, Norman, Oklahoma, United States of America, **5** Department of Pharmacy, Khwaja Yunus Ali University, Sirajganj, Bangladesh, **6** Department of Pharmacy, Mawlana Bhashani Science and Technology University, Tangail, Bangladesh, **7** Department of Pharmaceutical Chemistry, Bhupal Nobles' College of Pharmacy, Udaipur, Rajasthan, India, **8** Department of Chemistry, University of Rajshahi, Rajshahi, Bangladesh

* ahad.chembd@gmail.com

## Abstract

Cancer is characterized as a multifactorial disease due to their complex genetic and molecular mechanisms that often converge across tissue types. Shared oncogenic pathways can help us understand these functions and discover broad-spectrum therapeutics. Earlier, most studies focused on finding specific drivers for individual cancer types. However, researchers are now more interested in identifying common molecular patterns across different cancers and developing therapies that can target multiple pathways at once. This study aimed to understand the common oncogenic pathways between breast, ovarian and colorectal (BOC) cancers and identify possible multitargeted therapeutic drug molecules. To identify the common differentially expressed genes (DEGs), we analyzed three transcriptomic datasets and found a total of 128 DEGs. The protein-protein interaction (PPI) network study reveals the top-ranked, most significant hub targets, AURKA, CDK1 and CCNB1, as drug targets. Enrichment analysis with GO and KEGG pathways, as well as regulatory network (TFs and mRNAs) analysis, revealed common pathogenetic processes among BOC cancers. The AMG-900 exhibits the highest binding affinity scores of −10.8, −9.40, and −9.7 kcal/mol with the target proteins AURKA, CCNB1, and CDK1, respectively. The stability and structural flexibility of the selected protein-ligand complexes were validated by a large-scale (500 ns) molecular dynamics and MM-GBSA analyses, and the results indicate stable interactions for AURKA and CCNB1, while CDK1 showed comparatively reduced stability. The pharmacokinetic analysis revealed favorable drug-likeness and a manageable toxicity profile typical of anticancer agents.

**Data availability statement:** All relevant data are within the manuscript and its Supporting information files.

**Funding:** The author(s) received no specific funding for this work.

**Competing interests:** The authors have declared that no competing interests exist.

Therefore, the findings of this study propose that AMG-900 may serve as a promising multi-targeted candidate for further investigation in multi-target therapeutic strategies within precision oncology. Furthermore, these results require additional experimental (in vivo and in vitro) and clinical validation to confirm the potentiality and efficiency of this (AMG-900) lead compound.

## 1. Introduction

Cancer is a heterogeneous group of diseases, which characterized by abnormal cells proliferation, the ability to spread surrounding tissues, and potential metastasis to distant organs. Cancer is now commonly considered a multifactorial and systems-level disease rather than a defect in a single gene or pathway, due to its diverse genetic and epigenetic alterations converging on common signaling networks. Cancer is a major global threat to public health and ranks as the second leading cause of death in 112 of 183 countries [1–3]. Generally, damaged and aging cells are replaced by new ones through the cell cycle process. When this process is disrupted, uncontrolled cell growth can occur, which may lead to the development of either malignant or benign tumors [4,1]. In recent years, cancer treatment has gradually shifted toward more personalized approaches, where therapies are designed based on the specific genetic and molecular characteristics of a patient's tumor [5]. Targeted treatments, such as kinase inhibitors and monoclonal antibodies, often provide better outcomes with fewer side effects compared to traditional chemotherapy. At the same time, immunotherapy has opened new possibilities by enabling the immune system to recognize and attack cancer cells more effectively, with approaches like immune checkpoint inhibitors and CAR-T cell therapy showing promising results [6,7]. However, despite these advancements, issues such as drug resistance, tumor complexity, and treatment-related toxicity still remain significant challenges. This highlights the ongoing need to develop safer and more effective therapeutic strategies [8].

Breast cancer (BC), ovarian cancer (OC), and colorectal cancer (CRC) are significant malignancies, collectively representing a considerable share of global cancer incidence and mortality. Breast cancer (BC) is one of the most common cancers in women, with 16.67% deaths and almost 25% diagnosed, which makes it the fifth leading cause of cancer-related deaths globally [9]. Besides these statistics, it can be observed that about 1% of men are also affected by BC, and the increasing number of patients each year indicates a growing concern for global public health [10,11]. According to GLOBOCAN 2020, BC incidence rates are 88% higher in developed countries than developing countries, due to environmental factors, lifestyle choices (i.e., alcohol intake, ionizing radiation) and obesity [2]. Gynecologic cancers (GC), such as ovarian cancer (OC) is another deadliest cancer type in the present time, OC is often diagnosed at an advanced stage and has a poor prognosis due to its subtle onset, vague symptoms, and frequent diagnosis [12]. In 2020, OC ranked as the third most prevalent GC worldwide [13]. About 314,000 new cases and 207,000 deaths were attributed to OC in the same year, and the number was expected to rise over

the next years [2,14]. The risk factors for OC include gene mutations, a family history of the disease, and reproductive-related factors such as having children later or never having a full-term pregnancy, reaching menopause at an older age, and using hormone therapy after menopause [15]. In addition, colorectal cancer (CRC) is also one of the most prominent cancer types, making up about one-tenth of all cancer cases and deaths [2,16,17]. Though, the CRC is mostly observed in men, it also identified in women nowadays, usually within the age group of 65–74 [16]. A study conducted in 2020 reported that approximately 10% of colorectal cancer (CRC) cases were diagnosed, and nearly 9.4% of deaths were attributed to CRC compared to other cancer types. Experts predict that the number of new cases could reach 3.2 million as the global population increases by 2040 [18]. Like BC, lifestyle factors such as diet, physical inactivity, smoking, and obesity can increase the risk of colorectal cancer [16,19].

Although BC, OC, and CRC arise in different organs and have distinct clinical presentations, they share overlapping molecular alterations and oncogenic pathways that may be exploitable for multi-cancer therapeutic strategies. Although individual cancers have been well-studied, the molecular links between colorectal, breast, and ovarian cancers are not clear [20–22]. At the molecular level, several shared and distinct genetic mechanisms drive these cancers. For example, BRCA1 and BRCA2 play central roles in maintaining DNA stability through repair of double-strand breaks, and their mutations are found in 5–15% of breast and 10–25% of ovarian cancer cases [23–26]. In the meantime, colorectal cancer can be characterized by mutations in such key regulatory pathways as WNT, PI3K, TGF-β, and TP53. To illustrate, mutations in the APC gene have the potential of resulting in such a condition as familial adenomatous polyposis. The gene typically aids in the regulation of cell proliferation and tumors. It becomes defective which predisposes one to the development of CRC [27,28]. The knowledge of such molecular changes gives a background to the identification of multi-target therapeutic alternatives.

Thus, given the involvement of multiple genetic pathways across these cancers, therapeutic strategies focusing on a single molecular target may be insufficient. Because these cancers are controlled by complicated and often redundant signaling networks, treatments that focus on a single molecular target don't always work. They can be harmed by other pathways being activated and drug resistance developing quickly [29,30]. Single-target therapeutics are highly specific to a particular function; however, due to the redundant and adaptive nature of biological networks, they are often ineffective against complex diseases, leading to limited efficacy and rapid development of drug resistance [31]. On the other hand, multi-targeted drugs, which simultaneously modulate several disease-relevant pathways, have demonstrated enhanced therapeutic breadth, reduced heterogeneity and resistance, and greater treatment durability in multifactorial disorders [32]. For example, lapatinib is a dual EGFR/HER2 tyrosine kinase inhibitor used in HER2-positive breast cancer, and duvelisib is an oral dual PI3K-δ/γ inhibitor with activity in haematologic malignancies [33]. These clinical precedents support the concept that rationally designed or repurposed multi-kinase inhibitors could provide broader and more durable responses in cancers driven by multiple oncogenic drivers.

Recent advancements in computational approaches have reshaped how researchers design and optimize targeted therapeutics. The approach accelerates all processes involved in drug development including the process of identifying possible drug targets and screening compounds up to enhancing pharmacokinetic attributes and conducting clinical trials [34,35]. These types of computational methods are especially beneficial in the discovery of both the drug targets and candidate drug molecules that can act on a number of pathways related to cancer at the same time [36,37]. Furthermore, one of the most used methods of discovering new medications from existing drugs is known as drug repurposing (DR) [38,39]. There are numerous advantages of drug repurposing, which are: It is faster and cheaper to develop the drug, since much information about the safety, dosage, and side effects of the already developed drugs is available. Indicatively, the first drugs found to be effective in the treatment of Leukemia were chlorambucil and busulfone, [38,40–42] which were derived from mustard gas used during the world war-I [43]. In this context, integrative transcriptomic analyses combined with structure-based drug-repurposing strategies (molecular docking, molecular dynamics (MD) simulations, and ADMET) provide a powerful framework to systematically prioritize multi-target drug candidates for further preclinical evaluation.

Therefore, this study integrates a bioinformatic approaches to identify the common differentially expressed genes (DEGs) and common hub genes (cHGs) shared among breast, ovarian, and colorectal (BOC) cancers using publicly available transcriptomic (GEO) datasets. This aims to investigate and characterize the binding affinity and molecular interactions between approved or clinical-phase compounds and common hub genes (cHGs) identified across the selected cancers, using a range of bioinformatics tools and statistical analyses. Furthermore, molecular docking was performed, followed by post-docking MM-GBSA calculations and large-scale MD simulations to assess the stability and consistency of the protein–ligand complexes under different physiological conditions, thereby identifying the most promising candidate for drug repurposing. In addition, in silico ADME and toxicological evaluations were conducted to further assess the therapeutic potential and safety profiles of the selected compound.

## 2. Methods and materials

### 2.1. Data sources and descriptions

This section presents a systematic analysis of gene expression profiles from breast, colorectal, and ovarian cancer datasets to identify differentially expressed genes (DEGs) and determine the primary key hub proteins. These central molecular targets provide crucial insights into the underlying mechanisms and regulatory networks associated with cancer progression. The overall study plan given in the Fig 1.

**2.1.1. Data collection and microarray datasets.** In this study, we obtained gene expression data from three publicly available microarray datasets in the Gene Expression Omnibus (GEO) repository maintained by the National Center for Biotechnology Information (NCBI) (https://www.ncbi.nlm.nih.gov/geo/). The datasets GSE45827 (breast cancer),

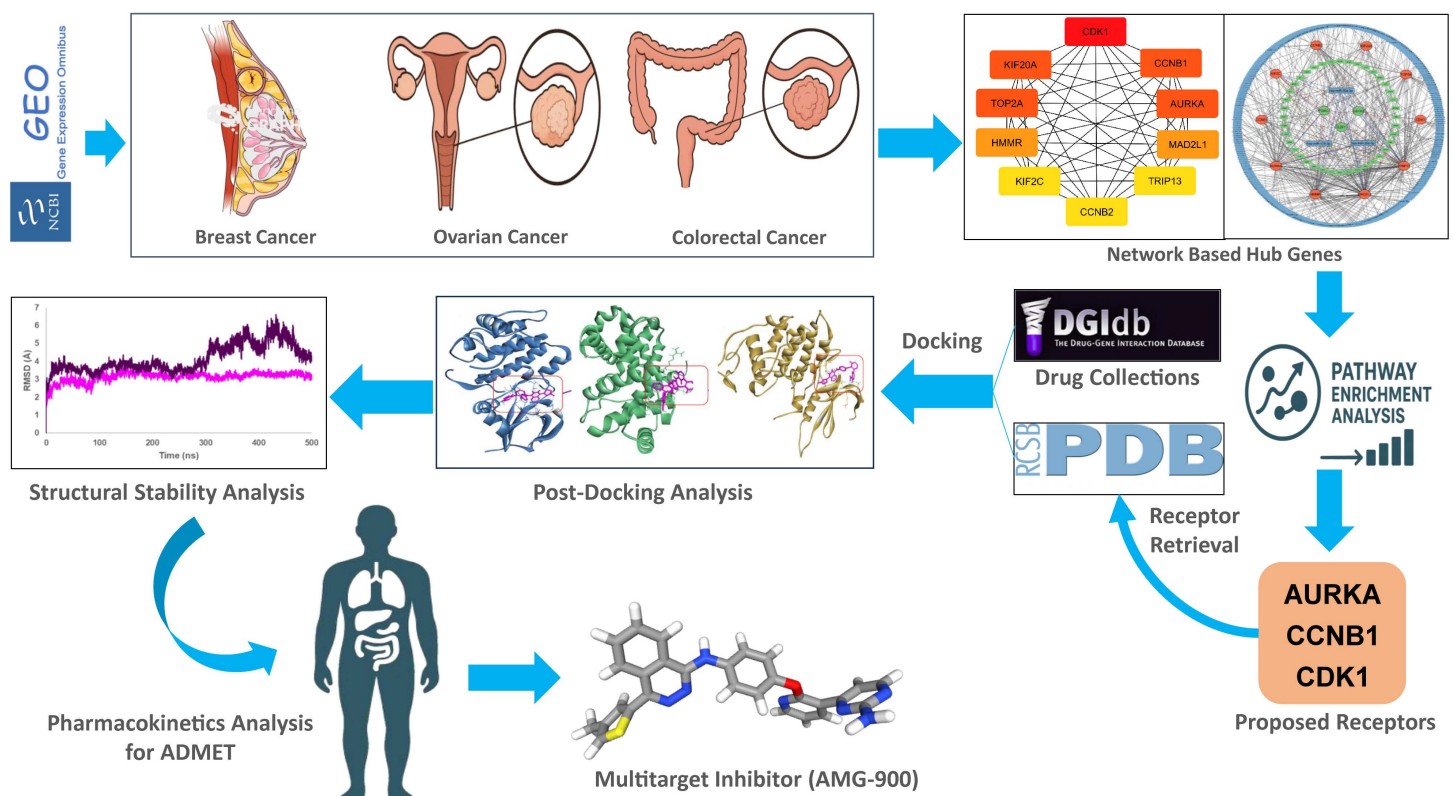

**Fig 1. Complete graphical representation/ working flowchart of this research.**

GSE21510 (colorectal cancer), and GSE26712 (ovarian cancer) contain transcriptional profiles from both tumor and corresponding normal tissue samples. Detailed information about each dataset, including the microarray platform and sample distribution, is provided in Table 1.

**2.1.2. Drug collection and ligand library preparation.** To identify potential inhibitors, a compound library was constructed using the Drug–Gene Interaction Database (DGIdb, https://www.dgidb.org) [48]. Approved and experimental drugs interacting with the three selected key hub proteins were retrieved and compiled. The corresponding chemical structures of these compounds were obtained from the PubChem database [49]. This approach generated a diverse set of small molecules suitable for subsequent virtual screening and molecular docking analyses to evaluate their binding potential and therapeutic relevance.

## 2.2. Identification of differentially expressed genes (DEGs)

To identify differentially expressed genes (DEGs) associated with the mentioned three cancers, we used NCBI GEO2R web tool (https://www.ncbi.nlm.nih.gov/geo/geo2r/) and we selected three different GEO datasets (GSE45827, GSE21510, and GSE26712) for each cancer type. In each dataset, samples were separated into two categories: cancer (case) and normal tissues (control). To normalize the expression data, it was pre-treated by log2 transformation. The linear models for microarray (LIMMA) package [50] and an empirical Bayes methodology was used to determine the differences between the expression of the genes between groups. Genes with an adjusted p-value less than 0.05 and a |logFC| greater than 1.0 were considered significantly differentially expressed. Positive logFC values indicated upregulation, while negative values indicated downregulation. Finally, the DEG lists of the three datasets were compared to determine common differentially expressed genes (cDEGs) among all the cancer types. The overlap of these sets of DEGs was identified with the help of the dplyr package [51] in R and the common genes were represented as Venn diagrams constructed with the help of the R packages ggplot2 [52] and ggvenn [53].

## 2.3. Protein–Protein interaction (PPI) network analysis

After identifying the cDEGs, we examined the possible interaction between the proteins encoded by these genes to identify the critical regulatory molecules of breast, colorectal, and ovarian cancers. The STRING database (https://string-db.org) was used to construct a protein-protein interaction (PPI) network with the focus on the experimentally validated interactions [54]. A minimum confidence score was established to 0.4, and only the query proteins were incorporated in order to select the direct association of the identified cDEGs. The interaction network obtained was plotted with Cytoscape (*version 3.10.4*) in which nodes are depicted by the proteins and the edges show the functional associations of these proteins [55]. In order to determine the most important nodes in the network, we used the CytoHubba plugin, which ranks nodes in terms of Degree in order to be robust in identifying them [56]. Proteins with a constant high score in this algorithm were identified as key hub proteins (KHPs) which represent the possible core regulators that can play crucial roles in the development and pathogenesis of such types of cancer.

**Table 1. Overview of the Publicly Available Datasets Used in This Study.**

| GEO Dataset | Types of Cancer | Number of Samples | Cancer | Control | Platform | Reference |
|---|---|---|---|---|---|---|
| GSE45827 | Breast cancer | 155 | 130 | 11 | GPL570 [HG-U133_Plus_2]; Affymetrix Human Genome; U133 Plus 2.0 Array | [44] |
| GSE21510 | Colorectal cancer | 148 | 123 | 25 | GPL570 [HG-U133_Plus_2]; Affymetrix Human Genome; U133 Plus 2.0 Array | [45] |
| GSE26712 | Ovarian cancer | 195 | 185 | 10 | GPL96[HG-U133A] Affymetrix Human; Genome U133A Array | [46,47] |

## 2.4. Analysis of transcriptional and post-transcriptional regulation of KHPs

We carried out an integrated regulatory network analysis that involves both transcriptional and post-transcriptional regulation to study the upstream regulatory mechanisms of the identified key hub proteins (KHPs). The results of transcription factor (TF)-target interaction were derived using the JASPAR database [57], and experimentally verified miRNA-target interaction using miRTarBase [58]. These datasets were imported into the NetworkAnalyst web platform (https://www.networkanalyst.ca/) that allows incorporating multi-layered molecular interactions and allows to perform topological evaluation of regulatory factors [59]. The TF-gene and miRNA-gene interaction networks were combined and interacted by Cytoscape (version 3.10.4). The regulatory factors displaying the highest connectivity and centrality rates were deemed important transcriptional or post-transcriptional regulating factors which may have a substantial influence on the expression and the functionality of the KHPs responsible of breast, colorectal and ovarian cancer pathogenesis.

## 2.5. Pathway enrichment analysis of KHPs

To clarify the biological role of the identified KHPs, the Gene Ontology (GO) and Kyoto Encyclopedia of Genes and Genomes (KEGG) pathway enrichment analyses were done with the help of the DAVID functional annotation tool (https://david.ncifcrf.gov/) [60]. The KHPs were divided by the GO analysis into three large groups, namely, biological process (BP), molecular function (MF), and cellular component (CC), and the highly enriched signaling pathways were identified by the KEGG analysis, both of which are directly retrieved by the DAVID. This enrichment analysis was done with default parameters of the DAVID database where the statistical significance cutoff was a p-value less than 0.05 and minimum number of genes were 2.

## 2.6. Validation of KHPs expression using GEPIA2 and UALCAN databases

The expression levels of the identified key hub proteins (KHPs) were further validated using publicly available cancer genomics and proteomics databases. Transcriptional validation was performed using the Gene Expression Profiling Interactive Analysis 2 (GEPIA2) web server (http://gepia2.cancer-pku.cn/), which integrates RNA sequencing data from The Cancer Genome Atlas (TCGA) and Genotype-Tissue Expression (GTEx) projects [61]. GEPIA2 was employed to compare the mRNA expression levels of selected hub genes between tumor and corresponding normal tissue samples. Boxplot analysis was generated using the default parameters of GEPIA2, with log2(TPM + 1) normalization, to visualize differential gene expression patterns across these three cancer types. To complement transcriptomic validation, protein-level expression analysis was conducted using the UALCAN database (https://ualcan.path.uab.edu/), which provides access to Clinical Proteomic Tumor Analysis Consortium (CPTAC) datasets [62]. The proteomics module of UALCAN was utilized to assess the relative protein abundance of the selected hub targets in cancer versus normal samples. Protein expression levels were statistically compared using Student's *t*-test as implemented by the platform, and the results were presented as boxplots.

## 2.7. Pan cancer analysis

### 2.7.1. Pan-cancer expression analysis using TIMER 2.0.
Pan-cancer expression analysis of the validated hub genes was conducted using the Tumor Immune Estimation Resource (TIMER 2.0) web server (http://timer.cistrome.org/). The Gene_DE module was employed to evaluate differential gene expression across multiple cancer types by comparing tumor and corresponding normal tissue samples from The Cancer Genome Atlas (TCGA). Gene expression differences were visualized using boxplots, and statistical significance was assessed using the Wilcoxon test implemented in TIMER 2.0 [63]. This analysis enabled the identification of additional cancers associated with the selected hub targets, supporting their broader oncogenic relevance.

 

**2.7.2. Immune infiltration analysis using TIMER 2.0.** The Immune-Gene module of the TIMER 2.0 platform was utilized to investigate the association between the selected hub targets and tumor immune cell infiltration. Among the available immune cell types, CD8+T cells were specifically analyzed due to their central role in anti-tumor immunity and their well-established involvement in cancer immune surveillance and immunotherapy response [64]. In particular, CD8+T cells are the primary effectors of anti-tumor immunity, and their infiltration level serves as a key prognostic biomarker and predictor of response to immunotherapies, including immune checkpoint inhibitors [65]. Therefore, focusing on CD8+T-cell infiltration provided biologically meaningful insights into the potential immunological relevance of the identified targets within the tumor microenvironment.

## 2.8. cKHPs-guided drug repurposing

**2.8.1. Retrieval and preparation of selected cKHPs.** The 3D crystal structure of selected cKHPs from the network analysis were retrieved from the UniProt [66] and RCSB Protein Data Bank (PDB) [67]. In this study, we mainly focused for the screening and selection of PDB IDs for the corresponding target protein based on several criteria including, the preparation methods (X-ray crystallography and NMR), resolution, presence of co-crystallized ligands (as positive control ligand), absence of mutation, and missing residues. In protein pre-processing step, the protein structures were refined by removing bound ligands, heteroatoms, and water molecules using BIOVIA Discovery Studio 2021 [68]. Each protein model was then subjected to energy minimization in Swiss-PDB Viewer (version 4.1.0) [69] to correct structural irregularities and ensure optimal stability for downstream molecular docking and simulation studies.

**2.8.2. Molecular docking study.** Molecular docking was conducted to investigate the interaction of the drugs with the three chosen key hub proteins. PyRx platform was used to dock all the collected compounds to the target proteins with a simple and intuitive interface for AutoDock Vina [70]. The grid box parameters used for each target protein are summarized in S1 Table. This setup allowed us to efficiently predict binding poses and estimate interaction strengths. In addition, here we docked the co-crystal ligand or the native ligand of the target protein structure to validate and ensure the docking protocol. To validate the docking protocol, the co-crystallized ligand was removed from the binding site and redocked into the prepared receptor using the same parameters applied for all test ligands. The agreement between the experimental and predicted poses was assessed by calculating the RMSD of ligand heavy atoms [71]. The best protein-drug complexes were then examined in detail using PyMOL [72] and BIOVIA Discovery Studio 2021, providing a clear view of the binding patterns and helping to understand which compounds might be most effective.

## 2.9. Molecular dynamics simulation study

To watch the behavior of the protein-drug complexes in a realistic physiological environment, we conducted simulations of molecular dynamics (MD) simulations [73]. Each complex was placed in a rectangular water box, leaving at least 10 Å of solvent around the protein. To mimic physiological conditions, the systems were neutralized with counterions and supplemented with Na+ and Cl− ions to reach an ionic strength of 0.15 M. The systems were first stabilized by the minimization of the steric clashes followed by a steady heating of the systems at 300 K under NVT conditions, after which an equilibrating process at 300 K under NPT conditions was performed to stabilize the density and pressure of the systems. Production simulations were performed over 500 ns with 2-fs time and OPLS3e force field [74], which also has better parameters on both biomolecules and drug-like ligands. The OPLS3e force field was selected for molecular dynamics simulations due to its high accuracy in modeling both protein and ligand systems, particularly in drug-like chemical space. It has been extensively parameterized and validated for ligan–protein interactions and is known to provide improved representation of torsional profiles, partial charges, and conformational energetics compared to earlier OPLS versions [74,75]. Any evaluation that occurs on one trajectory is well defined. Data of the simulation was then analyzed to determine the stability and flexibility of the complexes. RMSD, RMSF, SASA, radius of gyration, hydrogen bonds, and principal component analysis

(PCA) metrics have been used to monitor the structural variations and to examine the interaction of stabilizing the protein-ligand complexes.

## 2.10. Post simulation MM-GBSA calculation

To gain deeper insights into the energetics driving protein–ligand (PL) interactions, we estimated the binding free energy of each complex using the MMGBSA (Molecular Mechanics–Generalized Born Surface Area) approach. The analysis was carried out through the gmx_MMPBSA package, utilizing GROMACS trajectories that were first converted from Desmond simulation outputs. This method allowed us to evaluate how strongly the ligands interact with the target proteins beyond docking scores, providing a more reliable thermodynamic perspective.

For this study, MM/GBSA calculations were performed on the top three high-affinity ligands identified from docking and a reference compound to compare their binding tendencies. The free energy of binding ($\Delta G_{bind}$) was computed using the following relationship:

$$\Delta G_{bind} \; = \; \langle G_{PL} \rangle - \langle G_{P} \rangle - \langle G_{L} \rangle$$

Here, $\langle G_{PL} \rangle$, $\langle G_{P} \rangle$, and $\langle G_{L} \rangle$ denote the average free energies of the complex, the isolated protein, and the free ligand, respectively.

The total binding energy can be expressed as:

$$\Delta G_{bind} = \; \Delta E_{MM} + \; \Delta G_{SOLV} - \; T\Delta S$$

where $\Delta E_{MM}$ denotes the gas-phase molecular mechanics energy, including van der Waals and electrostatic interactions, $\Delta G_{SOLV}$ represents the change in solvation free energy, and T$\Delta S$ accounts for the entropic contribution. The polar solvation energy was estimated using the Generalized Born (GB) implicit solvent model, while the nonpolar term was evaluated using a solvent-accessible surface area (SASA) approach.

## 2.11. Pharmacokinetic properties evaluation

Following molecular docking and simulation analyses, the selected drug compound was further evaluated for their pharmacokinetic and toxicity profiles. The SwissADME web server (http://www.swissadme.ch/) was employed to predict key ADME parameters (absorption, distribution, metabolism, and excretion) and to assess drug-likeness, solubility, and oral bioavailability [76]. SwissADME provides comprehensive insights into molecular descriptors, physicochemical properties, and Lipinski's rule-based filters, helping identify compounds with favorable pharmacokinetic behavior.

To ensure the safety and therapeutic potential of this top-ranked compound, toxicity assessment was performed using the ProTox-III web server (https://tox-new.charite.de/protox_III/) [77]. This platform predicts various toxicity end-points, including hepatotoxicity, nephrotoxicity, cardiotoxicity, neurotoxicity and other clinical toxicity using advanced machine-learning models.

## 3. Result

### 3.1. Differential gene expression analysis

To investigate transcriptional changes in various cancers, we interrogated three independent GEO microarray datasets, namely GSE45827, GSE21510, and GSE26712, for breast cancer, colorectal cancer, and ovarian cancer, respectively. To compare the gene expression profile of each dataset, we considered the tumor and normal samples in our dataset. Following that, we performed normalization and statistical testing using the LIMMA package; the volcano plot (Fig 2A) illustrated the highly up- and downregulated genes in cancer tissue over controls. The differentially expressed genes

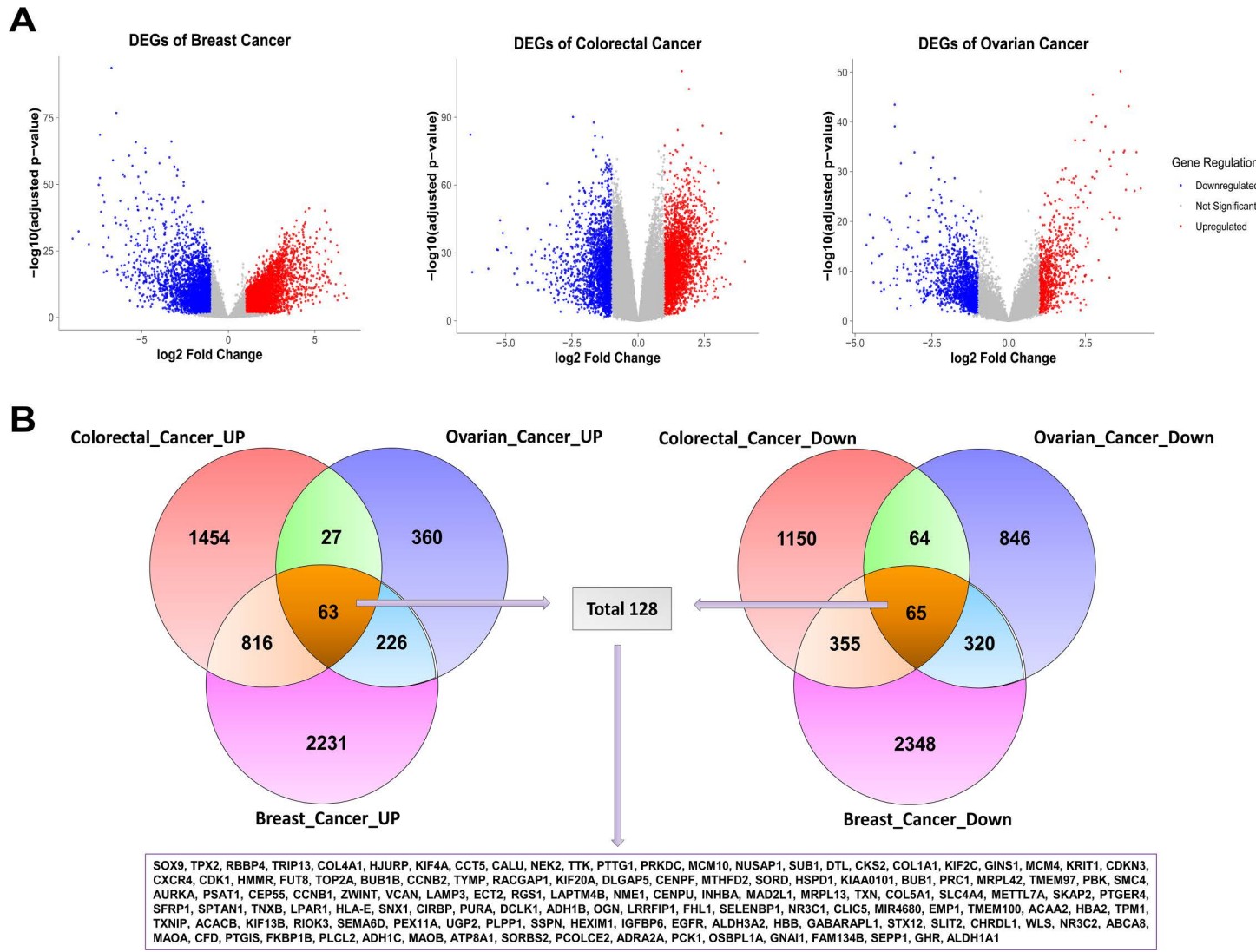

**Fig 2. Differential expression analysis and identification of shared DEGs among three cancer types.** (A) Volcano plots of DEGs in individual cancers and **(B)** Venn diagram showing overlapping upregulated and downregulated genes among breast, colorectal, and ovarian cancer datasets.

(DEGs) are listed in S2 Table as unique DEGs for three different datasets. In addition, to identify a set of common DEGs (cDEGs), we focused on genes that were consistently altered across all three cancers (Fig 2B), based on their shared oncogenic pathways for BOC cancers that typically have common pathways in cell cycle regulation, apoptosis, and signal transduction [78–80]. In this way, the identified cDEGs were considered fundamental molecular facilitators of tumor growth and were chosen to be further analyzed using the hub and discovered based on structures.

### 3.2. Identification of key hub-proteins (KHPs) using PPI network

The protein-protein interaction (PPI) network was performed using the STRING database to understand the interaction between identified cDEGs on the protein level. Experimentally validated interactions with a medium confidence score (0.4) were used to generate the network to obtain a balance between the reliability and the coverage of the data. The network

is represented using Cytoscape in Fig 3A, and every node represents a protein and the edge represents the functionality associations between the proteins.

In order to find out the strongest proteins that influenced the center of this interaction map, the Degree method of the CytoHubba plug-in in Cytoscape was utilized. This comparison pointed out the 10 hub proteins (CDK1, CCNB1, AURKA, KIF20A, TOP2A, HMMR, MAD2L1, KIF2C, TRIP13 and CCNB2) with the highest ranking (Fig 3B) as the most interesting because of their high connectivity and the possibility of being key regulators in the common cancer network. These central proteins likely coordinate multiple signaling pathways that drive tumor progression, making them promising targets for deeper molecular exploration and structure-based drug discovery. To further validate the robustness of these findings, a high-confidence threshold (0.7) was also applied, which preserved the core interaction clusters and retained key hub proteins, confirming that their identification is not dependent on low-confidence interactions (S1 Fig).

### 3.3. Analysis of transcriptional and post-transcriptional regulation of KHPs

The integrated regulatory network analysis revealed a complex interplay between transcriptional and post-transcriptional regulators influencing the expression of the identified key hub proteins. We identified three significant upstream regulators, that is, FOXC1, GATA2 and E2F1, which have strong connectivity and centrality in the network through interaction of TF-gene and interaction of miRNA-gene. Meanwhile, three microRNAs such as hsa-miR-92a-3p, hsa-miR-124-3p, and hsa-miR-20a-5p were identified as some of the important post-transcriptional regulators with a significant impact on various target proteins on various target proteins. These findings highlight a tightly coordinated regulatory framework potentially governing the molecular behavior of breast, colorectal, and ovarian cancers. The integrated TF–miRNA–gene interaction network illustrating these regulatory relationships is presented in Fig 4, which visually summarizes the central role of these factors in modulating KHPs expression and cancer progression.

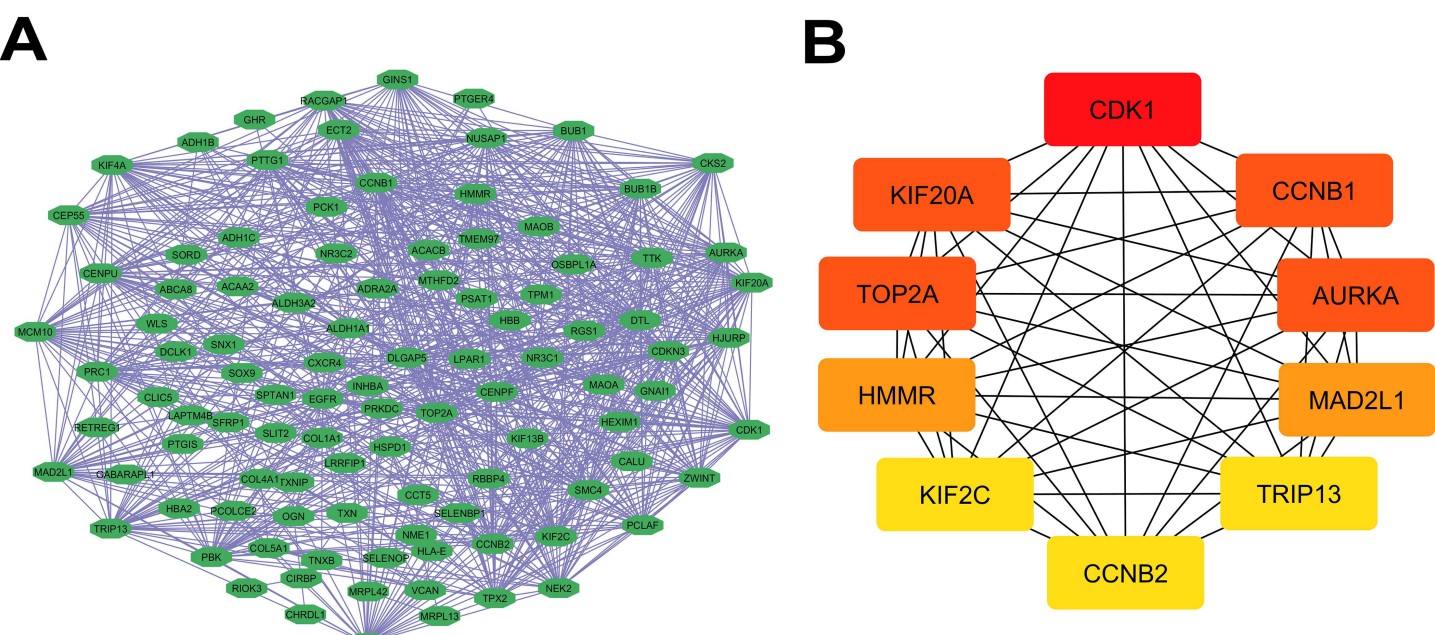

**Fig 3. Protein-Protein interaction (PPI) netwrok to identify KHPs from cDEGs. (A)** Construction of the PPI network from cDEGs and **(B)** identification of KHPs using Cytoscape.

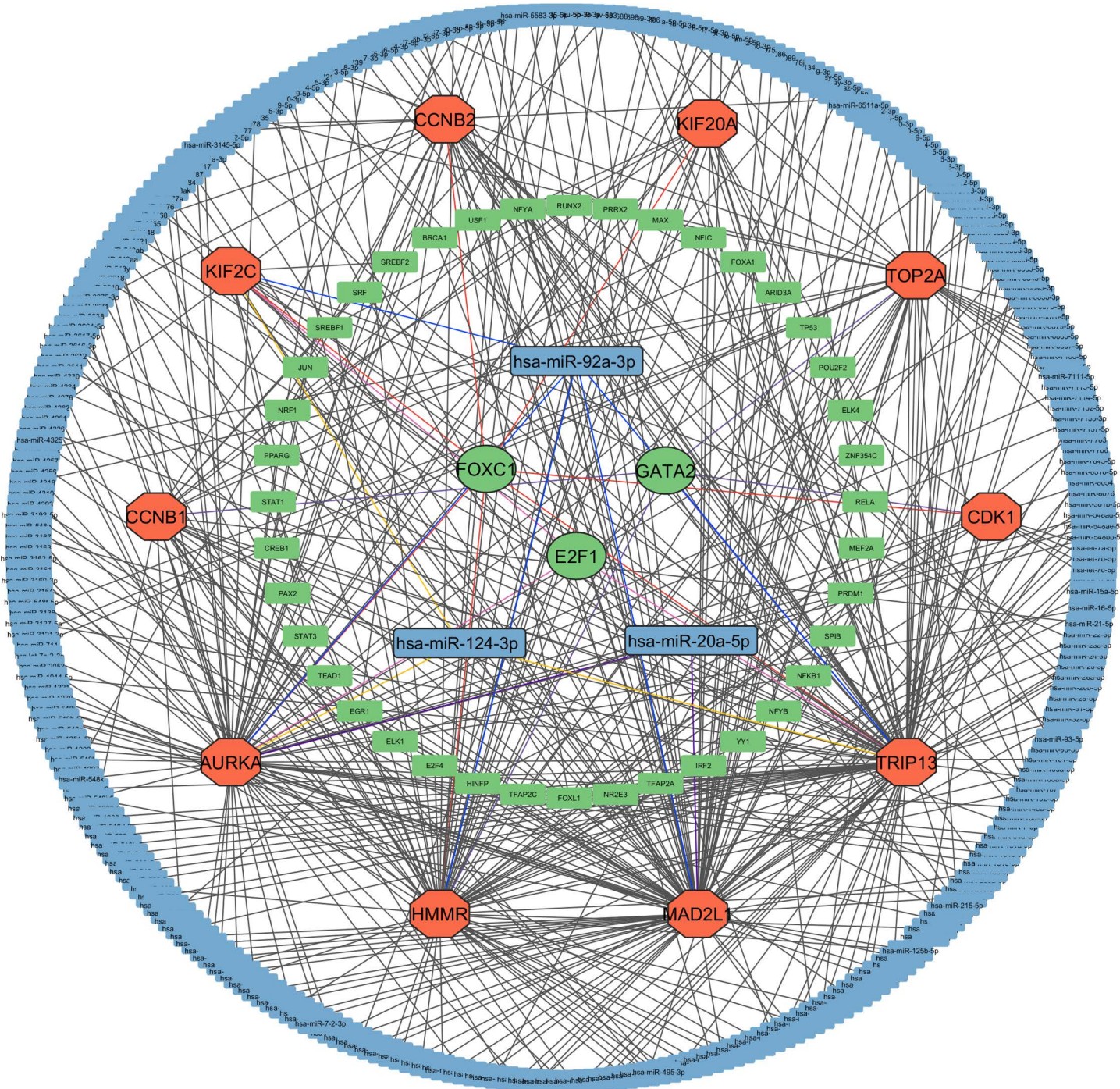

**Fig 4. Integrated TF–miRNA–gene regulatory network highlighting transcriptional and post-transcriptional regulators of key hub proteins across breast, colorectal, and ovarian cancers.** Green nodes represent transcription factors (TFs), orange nodes denote proteins, and sky-blue nodes indicate microRNAs (miRNAs).

### 3.4. GO and pathway enrichment analysis of KHPs

The enriched categories included biological processes (BP), cellular components (CC), molecular functions (MF) and signaling pathways, and detailed information in terms of the Table 2. The common hub genes, such as CCNB1, CCNB2, CDK1, and AURKA, are implicated in the enrichment pattern of the biological processes, which are vital in mitosis and cellular energy metabolism, which are crucial in cancer progression. Their consistent enrichment across breast, ovarian, and colorectal cancers points to a unified oncogenic mechanism characterized by disruptions in cell cycle control and instability during mitosis. This cellular component (CC) enrichment result showed that the shared hub genes as CCNB1, CDK1, KIF2C, and AURKA are predominantly localized in centrosome, mitotic spindle and nucleus. These elements are essential in proper segregation and division of chromosomes. Their high enrichment in the breast, ovarian and colorectal cancers implies that malformation of these subcellular structures can lead to errors in mitosis, and genomic instability to promote cancer development. According to the molecular function (MF) enrichment analysis, the common hub genes (CNB1, CCNB2, CDK1, AURKA, KIF2C, and KIF20A) are mostly involved with the binding of ATP, kinase activity, and the activity of the microtubules. Their functions are critical in the use of energy, transduction of signals and movement of chromosomes during the division of a cell. These genes have been steadily enriched in breast, ovarian and colorectal cancers indicating that a break in these molecular processes might be important in causing uncontrolled growth and tumor progression of cells. The KEGG pathway enrichment revealed that the common hub genes such as CCNB1, CCNB2, CDK1, AURKA, MAD2L1, and TRIP13 have a significant contribution to the cell cycle, oocyte meiosis, and the p53 signaling pathways. These signaling pathways are important in cell division, genomic stability and cell fate control. Their high enrichment in breast, ovarian, and colorectal cancers indicates that the dysregulation of these signaling cascades could be the cause of unregulated proliferation and cancer progression via defective cell cycle regulation and the loss of the response to DNA damage.

### 3.5. Validation of hub gene expression using GEPIA2 and UALCAN databases

The expression profiles of the selected hub genes (AURKA, CCNB1, and CDK1) were validated at both transcriptomic and proteomic levels using GEPIA2 and UALCAN databases across breast, ovarian, and colorectal cancers. GEPIA2-based mRNA expression analysis revealed that all three hub genes were significantly overexpressed in tumor tissues compared with corresponding normal samples across the analyzed cancer types, with all comparisons showing statistical significance ($p < 0.05$). These findings confirm the consistent transcriptional upregulation of AURKA, CCNB1, and CDK1 in breast, ovarian, and colorectal cancers (Fig 5A-5C).

   At the protein level, UALCAN proteomic analysis demonstrated cancer-type-specific expression patterns of the hub targets. CDK1 showed significant overexpression in both breast and colorectal cancer tissues compared with normal controls. AURKA exhibited significantly elevated protein expression in breast cancer; however, proteomic data for AURKA were not available for colorectal cancer in the UALCAN database. In contrast, CCNB1 displayed significant protein overexpression in breast and ovarian cancers. Collectively, these transcriptomic and proteomic validation results support the dysregulated expression of the identified hub genes across multiple cancers and reinforce their relevance as potential therapeutic targets (Fig 5a-5h).

### 3.6. Pan cancer analysis

   **3.6.1. Pan-cancer expression analysis of hub genes using TIMER 2.0.** Pan-cancer expression analysis using the Gene_DE module of TIMER 2.0 revealed widespread dysregulation of the validated hub genes across multiple cancer types. As illustrated in S2A Fig, AURKA showed significant differential expression in 18 cancer types compared with corresponding normal tissues. Among these, 16 cancer types exhibited highly significant expression changes ($p < 0.001$), while one cancer type showed significance at $p < 0.01$ and another at $p < 0.05$, indicating a strong and consistent pan-cancer association of AURKA. Similarly, CCNB1 exhibited significant differential expression in 17 cancer

**Table 2. Functional enrichment analysis of key hub proteins (KHPs).**

**Biological Process**

| GO ID | GO Term | Count | PValue | Genes |
|---|---|---|---|---|
| GO:0051301 | cell division | 6 | 4.04E-07 | CCNB2, CCNB1, CDK1, KIF2C, MAD2L1, AURKA |
| GO:0000086 | G2/M transition of mitotic cell cycle | 3 | 2.49E-04 | CCNB1, CDK1, AURKA |
| GO:0000082 | G1/S transition of mitotic cell cycle | 3 | 6.18E-04 | CCNB2, CCNB1, CDK1 |
| GO:0007057 | spindle assembly involved in female meiosis I | 2 | 0.001847 | CCNB2, AURKA |
| GO:1905448 | positive regulation of mitochondrial ATP synthesis coupled electron transport | 2 | 0.001847 | CCNB1, CDK1 |
| GO:0044772 | mitotic cell cycle phase transition | 2 | 0.00323 | CCNB2, CCNB1 |
| GO:0010971 | positive regulation of G2/M transition of mitotic cell cycle | 2 | 0.012866 | CCNB1, CDK1 |
| GO:0007094 | mitotic spindle assembly checkpoint signaling | 2 | 0.01378 | TRIP13, MAD2L1 |
| GO:0045931 | positive regulation of mitotic cell cycle | 2 | 0.01606 | CCNB1, AURKA |
| GO:0032465 | regulation of cytokinesis | 2 | 0.018335 | KIF20A, AURKA |
| GO:0007080 | mitotic metaphase chromosome alignment | 2 | 0.021966 | CCNB1, KIF2C |
| GO:0043066 | negative regulation of apoptotic process | 3 | 0.022208 | CDK1, MAD2L1, AURKA |
| GO:0007052 | mitotic spindle organization | 2 | 0.026036 | CCNB1, AURKA |
| GO:0042752 | regulation of circadian rhythm | 2 | 0.030542 | TOP2A, CDK1 |
| GO:0018105 | peptidyl-serine phosphorylation | 2 | 0.035028 | CDK1, AURKA |
| GO:0007018 | microtubule-based movement | 2 | 0.035923 | KIF2C, KIF20A |
| GO:0048511 | rhythmic process | 2 | 0.039496 | TOP2A, CDK1 |
| GO:0007059 | chromosome segregation | 2 | 0.045721 | TOP2A, KIF2C |
| GO:0001701 | in utero embryonic development | 2 | 0.098464 | CCNB2, CCNB1 |

**Cellular Component**

| GO ID | GO Term | Count | PValue | Genes |
|---|---|---|---|---|
| GO:0005813 | centrosome | 6 | 3.99E-06 | CCNB2, CCNB1, CDK1, KIF2C, HMMR, AURKA |
| GO:0072686 | mitotic spindle | 4 | 3.43E-05 | CDK1, KIF20A, MAD2L1, AURKA |
| GO:0005819 | spindle | 4 | 3.43E-05 | KIF2C, KIF20A, HMMR, AURKA |
| GO:0015630 | microtubule cytoskeleton | 4 | 1.32E-04 | CCNB2, KIF2C, HMMR, AURKA |
| GO:0005634 | nucleus | 9 | 4.35E-04 | TOP2A, CCNB2, CCNB1, CDK1, KIF2C, KIF20A, TRIP13, MAD2L1, AURKA |
| GO:0097125 | cyclin B1-CDK1 complex | 2 | 8.65E-04 | CCNB1, CDK1 |
| GO:0000922 | spindle pole | 3 | 0.001661 | CCNB1, MAD2L1, AURKA |
| GO:0000776 | kinetochore | 3 | 0.001994 | KIF2C, MAD2L1, AURKA |
| GO:0030496 | midbody | 3 | 0.003016 | CDK1, KIF20A, AURKA |
| GO:0005737 | cytoplasm | 8 | 0.003363 | TOP2A, CCNB2, CCNB1, CDK1, KIF2C, KIF20A, HMMR, MAD2L1 |
| GO:0005874 | microtubule | 3 | 0.008995 | KIF2C, KIF20A, AURKA |
| GO:0005829 | cytosol | 7 | 0.016216 | CCNB2, CCNB1, CDK1, KIF2C, HMMR, MAD2L1, AURKA |
| GO:0005876 | spindle microtubule | 2 | 0.016756 | CDK1, AURKA |
| GO:0005654 | nucleoplasm | 6 | 0.018545 | TOP2A, CCNB1, CDK1, KIF20A, MAD2L1, AURKA |
| GO:0000307 | cyclin-dependent protein kinase holoenzyme complex | 2 | 0.019737 | CCNB2, CDK1 |
| GO:0005871 | kinesin complex | 2 | 0.021437 | KIF2C, KIF20A |
| GO:0001673 | male germ cell nucleus | 2 | 0.024405 | TOP2A, TRIP13 |
| GO:0000775 | chromosome, centromeric region | 2 | 0.029054 | TOP2A, KIF2C |

*(Continued)*

**Table 2.** (Continued)

**Biological Process**

| GO ID | GO Term | Count | PValue | Genes |
|---|---|---|---|---|
| GO:0005815 | microtubule organizing center | 2 | 0.039545 | CCNB2, CCNB1 |
| GO:0005814 | centriole | 2 | 0.073657 | TOP2A, AURKA |

**Molecular Functions**

| GO ID | GO Term | Count | PValue | Genes |
|---|---|---|---|---|
| GO:0005524 | ATP binding | 6 | 3.18E-04 | TOP2A, CDK1, KIF2C, KIF20A, TRIP13, AURKA |
| GO:0016538 | cyclin-dependent protein serine/threonine kinase regulator activity | 2 | 0.014863 | CCNB2, CCNB1 |
| GO:0016887 | ATP hydrolysis activity | 3 | 0.016291 | KIF2C, KIF20A, TRIP13 |
| GO:0019901 | protein kinase binding | 3 | 0.021069 | CCNB1, KIF20A, AURKA |
| GO:0003777 | microtubule motor activity | 2 | 0.02725 | KIF2C, KIF20A |
| GO:0005515 | protein binding | 10 | 0.046432 | TOP2A, CCNB2, CCNB1, CDK1, KIF2C, KIF20A, TRIP13, HMMR, MAD2L1, AURKA |

**KEGG Pathway**

| KEGG IDs | KEGG Terms | Count | PValue | Genes |
|---|---|---|---|---|
| hsa04914 | Progesterone-mediated oocyte maturation | 5 | 3.25E-06 | CCNB2, CCNB1, CDK1, MAD2L1, AURKA |
| hsa04114 | Oocyte meiosis | 5 | 7.97E-06 | CCNB2, CCNB1, CDK1, MAD2L1, AURKA |
| hsa04110 | Cell cycle | 5 | 1.33E-05 | CCNB2, CCNB1, CDK1, TRIP13, MAD2L1 |
| hsa04115 | p53 signaling pathway | 3 | 0.002636 | CCNB2, CCNB1, CDK1 |
| hsa04218 | Cellular senescence | 3 | 0.011122 | CCNB2, CCNB1, CDK1 |
| hsa05170 | Human immunodeficiency virus 1 infection | 3 | 0.019885 | CCNB2, CCNB1, CDK1 |

types (S2B Fig), of which 16 cancers showed highly significant associations (p < 0.001) and one cancer type displayed significance at p < 0.01. In addition, CDK1 demonstrated significant overexpression in 16 different cancer types, with all observed associations reaching a high level of statistical significance (p < 0.001), as shown in S2C Fig. This uniform significance across multiple malignancies highlights the robust oncogenic involvement of CDK1.Collectively, these results confirm the broad pan-cancer relevance of AURKA, CDK1, and CCNB1 and further support their roles as common oncogenic drivers across diverse tumor contexts.

**3.6.2. Immune cell infiltration within CD8+ T-cell.** The relationship between the selected hub genes and CD8+T-cell infiltration was evaluated using the TIMER 2.0 platform, which integrates multiple immune deconvolution algorithms, including TIMER, EPIC, MCPCOUNTER, CIBERSORT and XCELL, to ensure robust and consistent estimation across different methods. As shown in S3a Fig, AURKA exhibited positive correlations with CD8+T-cell infiltration in thymoma (THYM), breast invasive carcinoma (BRCA), and uveal melanoma (UVM), suggesting a possible association between AURKA expression and cytotoxic T-cell presence within the tumor microenvironment. Similarly, CDK1 demonstrated positive correlations with CD8+T-cell infiltration in thymoma (THYM), head and neck squamous cell carcinoma (HNSC), breast invasive carcinoma (BRCA), and kidney renal clear cell carcinoma (KIRC), as illustrated in S3b Fig, indicating a potential, though not definitive, link with immune-related tumor features. For CCNB1, positive associations with CD8+T-cell infiltration were primarily observed in thymoma (THYM) and lung adenocarcinoma (LUAD), as presented in S3c Fig. Overall, these correlations were modest and should be interpreted with caution, highlighting a tentative relationship between hub gene expression and CD8+T-cell infiltration across selected cancer types. While these findings provide preliminary insights, they do not establish a direct mechanistic connection, and further experimental and clinical validation is necessary to clarify any potential immunological or therapeutic relevance.

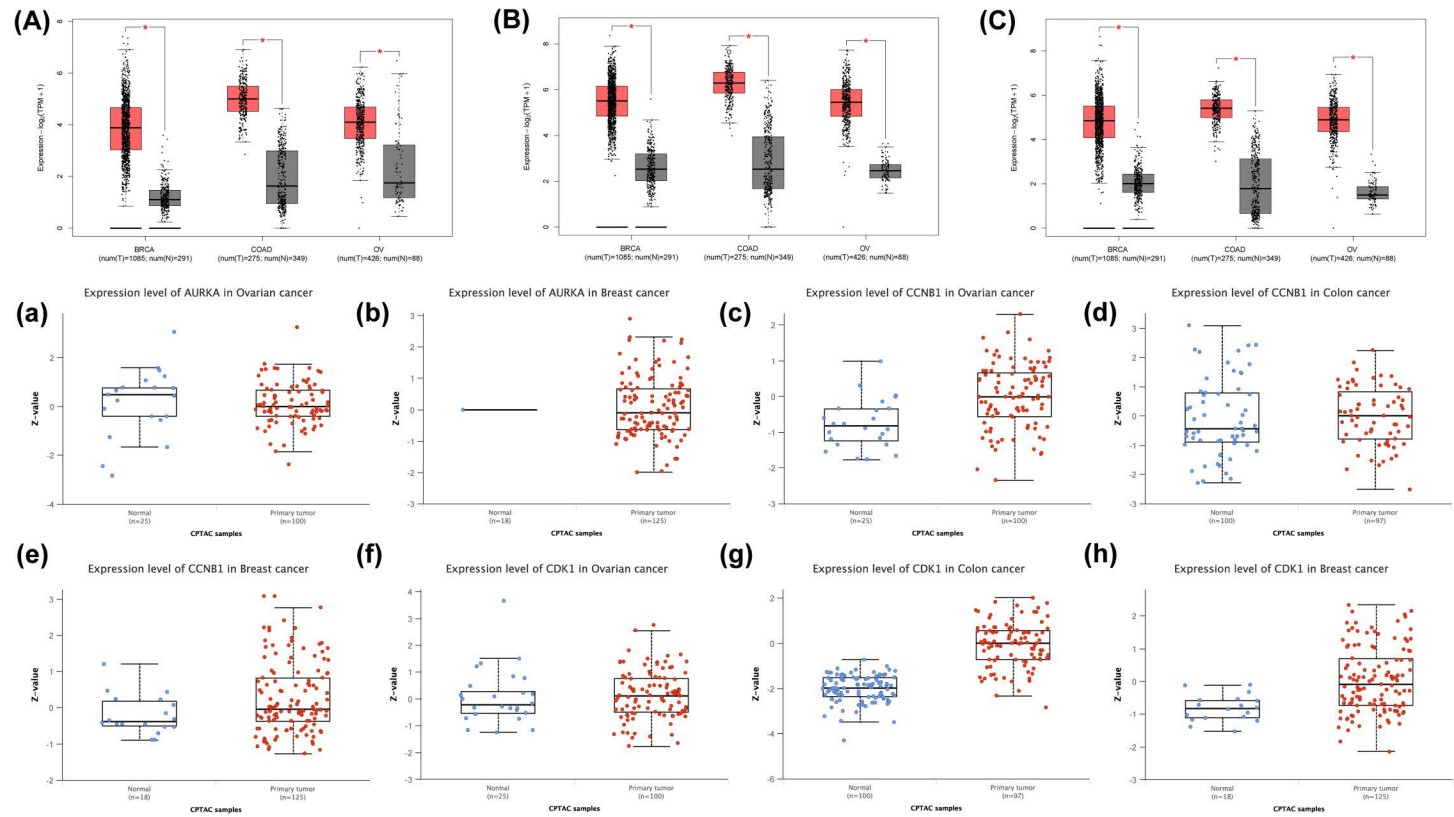

**Fig 5. Validation of potential therapeutic drug targets through box plot analysis using GEPIA2 (mRNA, A–C) and UALCAN (Protein, a–h).**

## 3.7. Molecular docking study

The crystal structures of the selected top-ranked target proteins AURKA, CCNB1, and CDK1 were retrieved using their corresponding PDB IDs 3HA6, 4Y72, and 6GU6 respectively. These structures were selected based on their high resolution, presence of co-crystallized ligands, and absence of mutations, making them suitable for reliable docking. To ensure structural accuracy and stability, each protein model was preprocessed by eliminating bound water molecules and non-essential heteroatoms. A total of 107 potential meta-drugs were collected from DGIdb database. Then, the docking was performed to calculate the binding interaction between the collected drugs (ligands) and proposed receptor protein (drug targets). A complete list of binding affinity scores (BAS) in kacl/mol for all protein–ligands are provided in S3 Table and the top-ranked 20 compounds based on their average highest BAS, ranging from −8.5 to −9.9 kcal/mol, and their corresponding scores are presented in Fig 6. From the listed compounds we then select AMG-900 (−9.96 kcal/mol) as the highest scorer multi-targeted drug candidates against our proposed receptor proteins.

Further, to ensure whether the docking protocol was appropriate for we performed re-docking with the native ligand 2JZ, LZ9, and 1Qk for AURKA, CCNB1, and CDK1 respectively. These ligands are experimentally validated inhibitors of the selected target protein and also work as a positive control for such study [81]. The docking score of 2JZ (−9.5 kcal/mol), LZ9 (−8.88 kcal/mol), and 1Qk (−8.5 kcal/mol) indicating good binding interaction with the target protein. While, our proposed AMG-900 showed the average binding affinity −9.96 kcal/mol (−10.80, −9.40, and −9.70 kcal/mol respectively) suggest highest binding affinity towards the target proteins than the positive co-crystal ligand. Additionally, The best-scored redocked pose of the native ligands (2JZ, LZ9, and 1Qk) showed an RMSD of 1.649, 2.858, and 0.825 Å

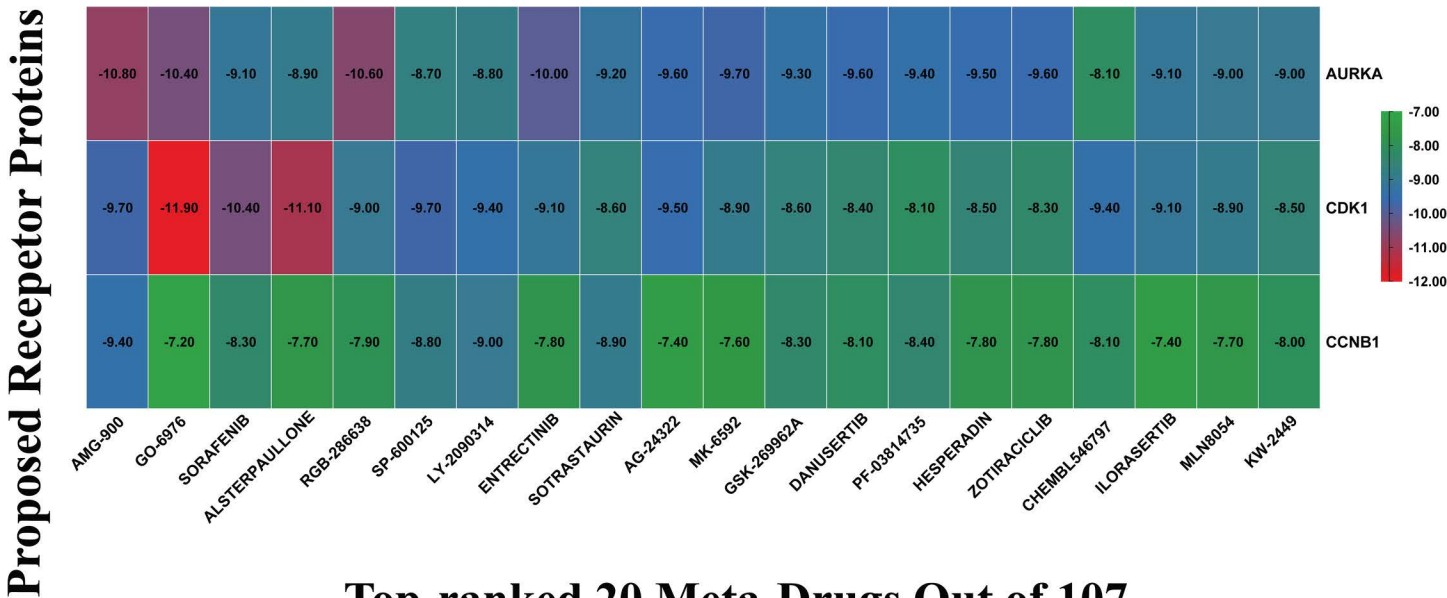

**Fig 6. Graphical Presentation of drug-target binding affinity score matrices (kcal/mol), where the X-axis indicates top-ordered 20 meta-drug (out of 107) and Y-axis indicates top-ordered proposed receptor proteins AURKA, CCNB1, and CDK1.**

respectively (S4 Fig), relative to the crystal structure of the selected structure, which falls within the commonly accepted cutoff (≤ 2.0 Å) for accurate pose prediction, supporting the robustness of the established docking settings [82,83].

### 3.8. Protein–Ligand molecular interactions

Among all screened compounds, AMG-900 was identified as the top ligand exhibiting strong and stable interactions with all three target proteins (AURKA, CCNB1, and CDK1). This compound demonstrated the most favorable binding profiles across the docking studies, indicating its potential as a multi-target inhibitor. Specifically, the AMG-900 strong binding to AURKA and CCNB1 by forming two covalent-hydrogen bonds with a docking score of –10.80 and –9.70 kcal/mol respectively, and the complex of CDK1 with AMG-900 demonstrated four hydrogen bonds with a docking score of –9.40 kcal/mol. These results indicate a strong and stable binding tendency of AMG-900 toward the active sites of the selected proteins. The detailed information on hydrogen bonding and hydrophobic interactions for each protein–ligand complex is provided in S4 Table, while the corresponding 2D and 3D interaction visualizations are illustrated in Fig 7.

### 3.9. Molecular dynamics simulation

A 500 ns (0.5 µs) molecular dynamics (MD) simulation was conducted to determine the stability and conformational dynamics of the protein-ligand complex in physiological conditions. The simulation helped us to observe the variability of structure and stability of interaction over a longer time scale. All reported MD metrics, including RMSD, RMSF, radius of gyration (Rg), and solvent-accessible surface area (SASA), represent the averaged results (mean±SEM) obtained from three independent simulation runs to ensure statistical reliability (Table 3). The use of SEM provides an estimate of the variability between independent simulations and reflects the precision and reliability of the calculated mean values. A lower SEM indicates greater consistency among the simulation runs, strengthening confidence in the reproducibility of the results. Thus, reporting mean±SEM ensures a more statistically robust and interpretable representation of the MD

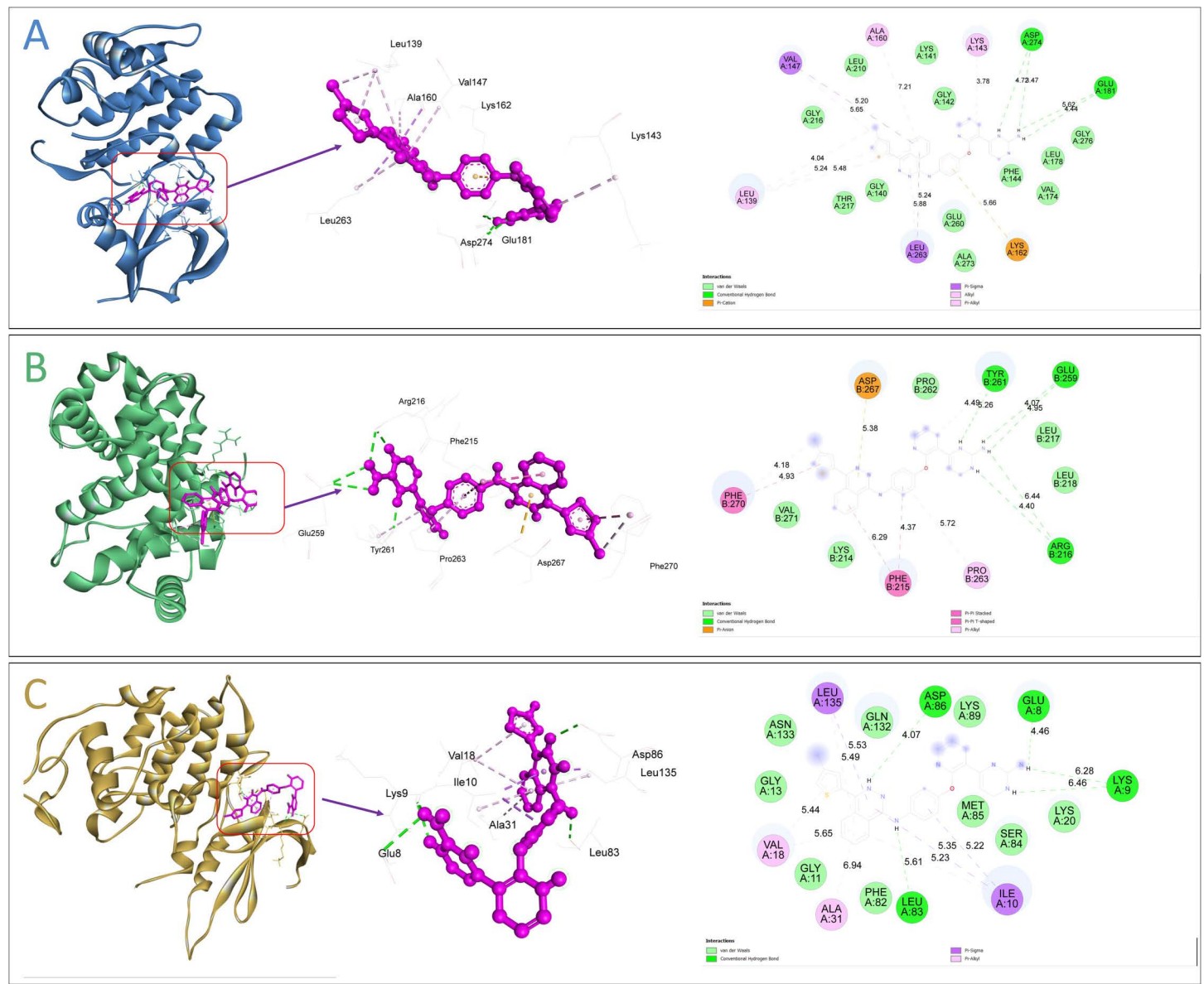

**Fig 7. 3D and 2D interaction representations using BIOVIA Discovery Studio of AMG-900 docked with (A) AURKA, (B) CCNB1, and (C) CDK1 proteins, showing key hydrogen bonding and hydrophobic interactions within the binding sites.**

simulation data. Besides, the principal component analysis and dynamical cross-correlation matrix were analyzed along the simulation path to investigate structural conformations and molecular motions in a better way.

**3.9.1. Root-mean-square deviation analysis.** Root-mean-square deviation (RMSD) analysis was done to determine whether there was any conformational change in the structure [84]. When the value of RMSD is of more than 3 Å, a considerable conformational change of the protein structure is usually implied. In order to determine the structural aberrations with respect to the apo form, the RMSD values of the lead compound AMG-900 have been calculated and are shown in Fig 8A. The RMSD average of the three proteins AURKA, CCNB1, and CDK1 were 3.07, 3.21, and 3.14 Å, respectively. Upon binding with the lead compound AMG-900, the values of their RMSD became 2.77, 2.79, and 4.15

**Table 3. Molecular dynamics simulation parameters and MM-GBSA binding free energy calculations for the protein–ligand complexes. The values within the first bracket (.) indicate the standard error of the mean (SEM).**

| Compounds Name | RMSD (± SEM) (Å) | RMSF (± SEM) (Å) | SASA (± SEM) (Å²) | Rg (± SEM) (Å) | H-Bonds (± SEM) | MM-GBSA (± SEM) (kcal/mol⁻¹) |
|---|---|---|---|---|---|---|
| Apo (AURKA) | 3.07 (0.006) | 1.56 (0.07) | – | – | – | – |
| Apo (CCNB1) | 3.21 (0.005) | 1.54 (0.06) | – | – | – | – |
| Apo (CDK1) | 3.14 (0.03) | 1.31 (0.05) | – | – | – | – |
| AURKA- AMG-900 | 2.77 (0.004) | 1.41 (0.06) | 288.48 (1.03) | 5.88 (0.001) | 0.256 (0.006) | −14.50 (4.56) |
| CCNB1- AMG-900 | 2.79 (0.003) | 1.35 (0.03) | 359.13 (0.47) | 5.94 (0.002) | 0.006 (0.001) | −11.99 (0.17) |
| CDK1- AMG-900 | 4.15 (0.01) | 2.07 (0.07) | 432.08 (1.30) | 5.87 (0.003) | 0.019 (0.001) | 3.03 (0.12) |

**Fig 8. RMSD and RMSF analyses of AURKA, CCNB1, and CDK1 in apo and AMG-900 bound states showing structural stability and residue flexibility changes upon ligand binding.**

Å, respectively. Reduced RMSD of AURKA and CCNB1 would indicate an improved structural stability where the two proteins have adopted more compact and high-rigid structures that allow binding of the ligand in a more stable manner. Conversely, the RMSD of CDK1 was higher, which suggests that a significant change in conformational arrangement happened probably as a result of allosteric effects or domain motions. Despite the fact that such a deviation reflects high ligand binding, it also could be taken to imply that the CDK1 complex was more flexible or destabilized to a small degree, so further research is required to elucidate its functional implications.

**3.9.2. Root-mean-square fluctuation analysis.** The root-mean-square fluctuation (RMSF) measures how much each atom fluctuates around its average position throughout the simulation, providing insight into the flexibility of individual residues [85]. The RMSF values of both the apoproteins and their complexes with the selected compound (AMG-900) were analyzed to assess how the interactions affected protein flexibility at specific residue positions (Fig 8B). On average, the AURKA protein showed RMSF values of 1.566 Å in its apo form and 1.419 Å when bound to the compound, while CCNB1 displayed 1.546 Å and 1.354 Å, respectively. In contrast, CDK1 exhibited a slight increase in flexibility upon

binding, with RMSF values rising from 1.314 Å in the apo form to 2.072 Å in the complex. For AURKA and CCNB1, the decrease in RMSF values upon ligand binding indicates that the compound stabilizes these proteins by reducing fluctuations in flexible loop regions, such as residues 9–172 in AURKA, where the peaks appear more confined in the complex form. This reduction in flexibility suggests that the ligand helps rigidify dynamic regions by occupying the active site and limiting conformational movements, which is often a characteristic of strong kinase inhibitors. In contrast, CDK1 exhibited an increase in RMSF upon complex formation, particularly in the N-terminal nucleotide-binding area (residues 40–43, just before the αC-helix), with subtler shifts in the activation loop (around residues 146–173). Overall, these findings indicate that AMG-900 effectively stabilizes AURKA and CCNB1, making them promising therapeutic targets, while its interaction with CDK1 appears less favorable, highlighting the need for further optimization to improve selectivity.

### 3.9.3. Solvent accessible surface area.

The solvent-accessible surface area (SASA) is the area of the surface of a protein that is open to the point of contact with the surrounding solvent molecules. It is an important part of the explanation of protein folding and stability [86]. SASA can be determined by the trajectory of the center of a solvent molecule rolling over the surface of a protein and it takes into consideration van der Waals interactions [87]. Proteins typically have amino acid residues on the outer surface which are active sites and hence are capable of binding with ligands and other structures. The study of SASA aids in the understanding of how proteins and ligands are going to interact under different conditions, such as hydrophobic and hydrophilic solvents [88]. In this study, the SASA profiles of AURKA, CCNB1, and CDK1 complexes with AMG-900 revealed average solvent-accessible surface areas of 288.48, 359.13, and 432.08 Å², respectively, over a 500 ns simulation (Fig 9A). The relatively compact SASA observed for the AURKA complex suggests tight ligand binding and conformational stability. In contrast, the slightly higher SASA values in CCNB1 and CDK1 reflect maintained structural flexibility, which supports dynamic stability and efficient ligand accommodation. Taken altogether, the steady trends of the SASA among all the three complexes show that there is structural strength and positive adaptability during the simulation phase.

### 3.9.4. Radius of gyration.

The radius of gyration (Rg) analysis was conducted to determine the compactness and stability of the protein-ligand complexes in the course of the 500 ns simulation (Fig 9B). It is an important measure of molecular compactness, where higher Rg values indicate a more expanded or flexible structure, while lower values suggest a tighter and more compact conformation [89]. The average Rg values for AURKA-AMG-900, CCNB1-AMG-900, and CDK1-AMG-900 were 5.88, 5.94, and 5.87 Å, respectively, all showing consistent stability throughout the simulation. AURKA and CCNB1 complexes displayed steady fluctuations (std. dev. 0.11 and 0.17 Å), while CDK1 showed slightly higher variability (std. dev. 0.22 Å) but remained structurally stable overall. These findings indicate that the binding to AMG-900 does not produce significant conformational alterations, which means that each of the three complexes had compact and stable structures in these conditions.

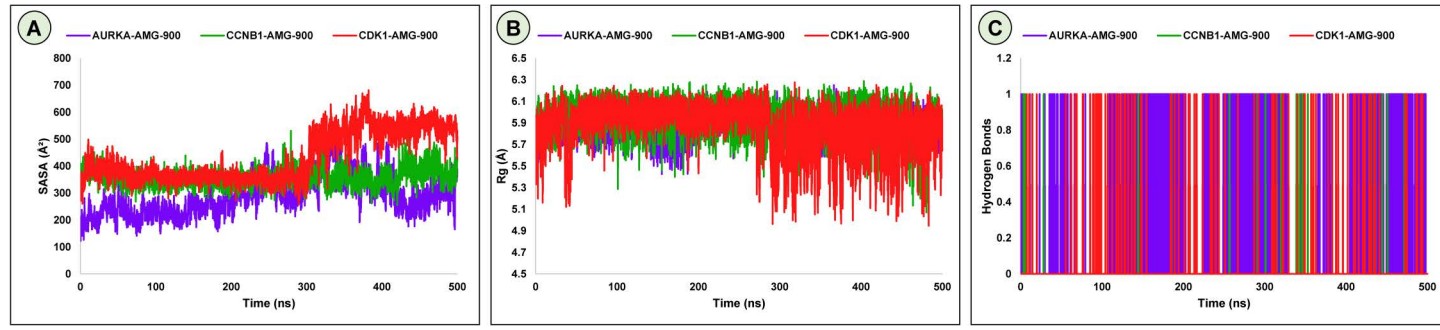

**Fig 9. SASA (A), Rg (B) and Hydrogen Bond (C) analyses of AURKA, CCNB1, and CDK1 complexes with AMG-900 showing solvent exposure and structural compactness during 500 ns simulation.**

**3.9.5. Hydrogen bond analysis.** Hydrogen bonds play a crucial role in stabilizing ligand–protein interactions and greatly influence a drug's specificity, metabolism, and absorption [90]. Among the three complexes, AURKA- AMG-900 exhibited the highest average hydrogen bond occupancy (0.256), indicating strong and persistent interactions throughout the simulation. This suggests superior binding stability and reduced conformational fluctuations. CCNB1 (0.006) and CDK1 (0.019) also maintained consistent but comparatively weaker hydrogen bonding, reflecting moderate yet stable ligand engagement. Overall, all three complexes demonstrated stable hydrogen bond profiles, with AURKA showing the most pronounced affinity, followed closely by CDK1 and CCNB1, supporting the compound's effective and selective binding within the AURKA-CCNB1-CDK1 regulatory network (Fig 9C).

## 3.10. MM-GBSA-based binding free energy calculation

The Molecular Mechanics Generalized Born Surface Area (MM-GBSA) binding free energy (BFE) analysis was performed to understand how strongly the ligand binds with the target proteins AURKA, CCNB1, and CDK1 during the 500 ns MD simulation (Fig 10). The average BFEs were found to be −14.50 kcal/mol for AURKA, −11.99 kcal/mol for CCNB1, and +3.03 kcal/mol for CDK1. A more negative value generally means a stronger and more stable interaction between the ligand and protein [91]. Based on this, the AURKA and CCNB1 complexes showed favorable and stable binding, suggesting that AMG-900 interacted well with these proteins. However, CDK1 exhibited a positive binding free energy, indicating

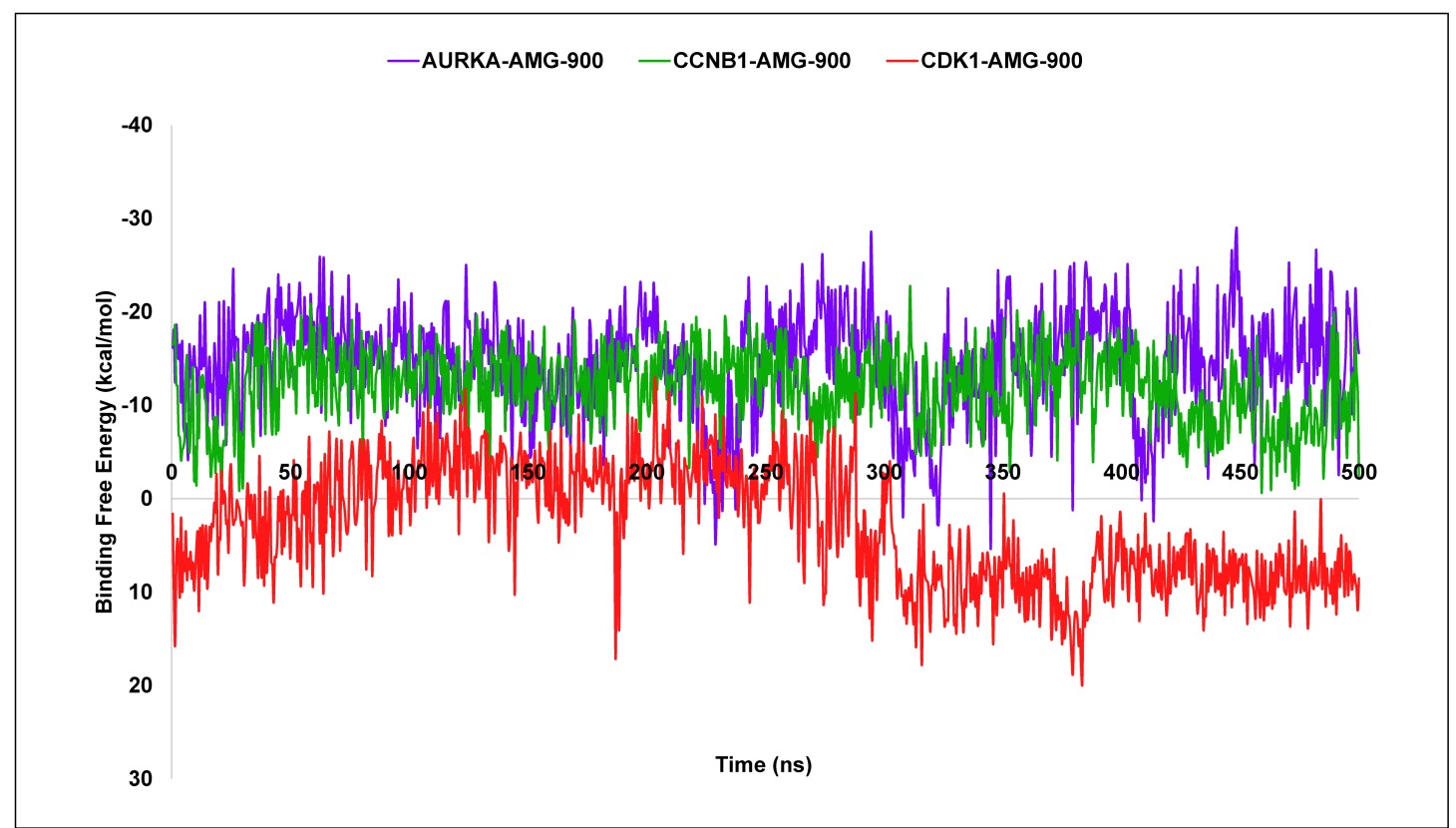

**Fig 10. MM-GBSA binding free energy analyses of AURKA, CCNB1, and CDK1 complexes with AMG-900 showing interaction stability and binding strength.**

that the interaction is thermodynamically unfavorable. This suggests that the CDK1–AMG-900 complex may not form a stable inhibitory interaction under dynamic conditions.

### 3.11.  Principal component analysis

Principal component analysis (PCA) of the 500-ns trajectories was performed to explore the dominant collective motions of the three complexes [92]. The first three principal components (PC1-PC3) captured most of the essential atomic fluctuations in each system. For the CDK1 complex, PC1 accounted for 53.68% of the variance (PC2: 12.48%, PC3: 3.97%; cumulative 70.1%), indicating a dominant coordinated motion. The compact clustering in the PC1-PC2 plot suggests that AMG-900 effectively restricted large-scale fluctuations, enhancing conformational stability. The AURKA-AMG-900 complex also showed stable dynamics (PC1: 45.08%, PC2: 8.98%, PC3: 5.96%), with gradual convergence of trajectories over time, reflecting consistent stabilization under ligand binding. In comparison, the CCNB1-CID 24856041 complex displayed slightly broader conformational sampling (PC1: 28.16%, PC2: 12.11%, PC3: 9.59%), indicating that while the protein retained some flexibility, the overall motions were still confined within a stable conformational space. As illustrated in Fig 11, AMG-900 contributes to maintaining structural stability across all three

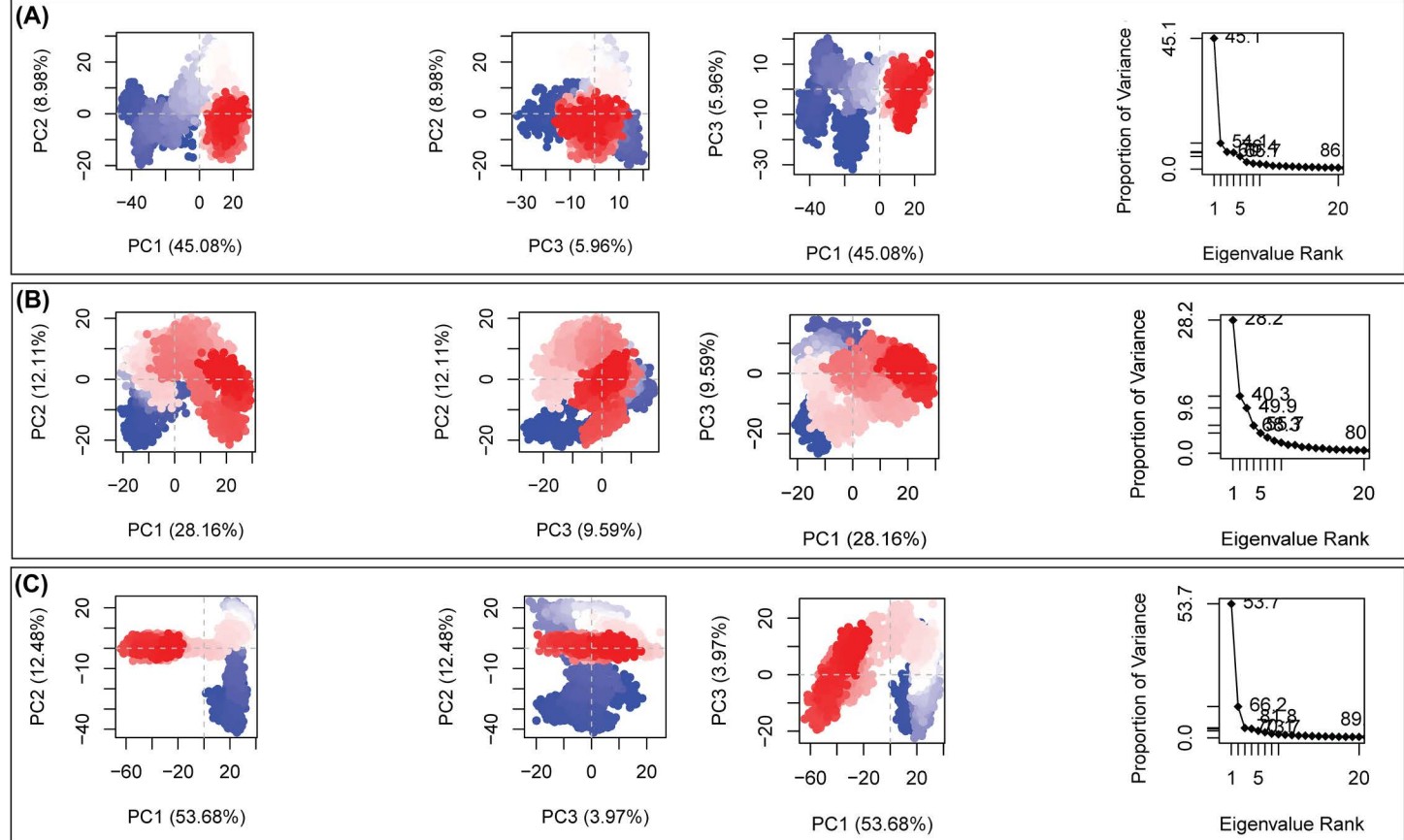

**Fig 11.  Principal Component Analysis (PCA) plots showing the dynamic behavior of the bound protein 6GU6 with the ligands, where (A) AURKA_AMG-900 complexes, (B) CCNB1_AMG-900, and (C) CDK1_AMG-900.** Color mapping further clarified the fluctuation pattern across structures, with the red regions representing the least dynamic fluctuations (stable region), the white zones indicating moderate movements, and the blue areas highlighting the highest conformational change (dynamic region).

complexes, with a particularly pronounced stabilizing effect on CDK1 and AURKA. The ligand's ability to modulate dynamic behavior in multiple targets underscores its potential as a versatile multi-kinase inhibitor for cancer therapy.

### 3.12. Pharmacokinetics study

**3.12.1. Prediction of physicochemical and ADME properties.** The ADME profile of AMG-900 was summarized in Table 4. The compound showed moderate drug-likeness, with one Lipinski rule violation due to its slightly high molecular weight. The compound had a bioavailability score of 0.55, suggesting it might have faced some challenges with oral absorption. However, it showed no PAINS or Brenk alerts, indicating a low risk of false-positive biological activities, and it also had good synthetic accessibility, meaning it could be synthesized without major difficulty. The compound was highly lipophilic and rather rigid, which contributed to its poor water solubility in all prediction models. As a result, it showed low gastrointestinal absorption and was unable to cross the blood–brain barrier. Its relatively large polar surface area and low skin permeability further limited its distribution in the body. Importantly, it was not a substrate of P-glycoprotein, suggesting it would not have been actively pumped out of cells. From a metabolic perspective, AMG-900 acted as a CYP3A4 inhibitor but remained inactive against other key cytochrome P450 enzymes, implying that it would not have strongly interfered with most metabolic pathways in the liver.

**3.12.2. Toxicity analysis.** The toxicity profile, predicted using ProTox-3.0, was presented in Table 5. AMG-900 demonstrated an overall favorable safety margin for early-stage drug development. The compound showed no signs of kidney, heart, or general cellular toxicity, and remained inactive in most nuclear receptor and stress response pathways, indicating a low risk of hormonal disruption or oxidative stress. It also exhibited minimal interference with major drug-metabolizing enzymes, implying stable metabolic behavior. While some potential for hepatic, neurological, and respiratory effects was predicted, this was expected for anticancer agents and remained within tolerable limits for a lead compound,

**Table 4. Predicted ADME properties of AMG-900 obtained from SwissADME web server.**

| Properties | Property | Value |
|---|---|---|
| Physicochemical | Molecular formula | $C_{28}H_{21}N_7OS$ |
| | Molecular weight (Da) | 503.58 |
| | Number of rotatable bonds | 6 |
| | Number of H-bond acceptors | 6 |
| | Number of H-bond donors | 2 |
| | Log P | 4.83 |
| | Topological polar surface area (Å²) | 139.97 |
| Pharmacokinetics | Solubility Log S | −6.44 |
| | Gastrointestinal (GI) absorption | Low |
| | P-glycoprotein substrate | No |
| | Blood-brain barrier permeant | No |
| | CYP1A2 inhibitor | No |
| | CYP2C19 inhibitor | No |
| | CYP2C9 inhibitor | No |
| | CYP2D6 inhibitor | No |
| | CYP3A4 inhibitor | Yes |
| | Skin permeability (log K_p, cm/s) | −5.75 |
| Medicinal Chemistry | PAINS alerts | 0 |
| | Brenk alerts | 0 |
| | Lead-likeness violations | 2 |
| | Synthetic accessibility score | 3.96 |

**Table 5. Predicted toxicity profile of AMG-900 based on ProTox-3.0 analysis.**

| Target | Prediction | Probability | Target | Prediction | Probability |
|---|---|---|---|---|---|
| Hepatotoxicity | Active | 0.6 | Aryl hydrocarbon Receptor (AhR) | Active | 0.75 |
| Neurotoxicity | Active | 0.7 | Androgen Receptor (AR) | Inactive | 0.87 |
| Nephrotoxicity | Inactive | 0.62 | Androgen Receptor Ligand Binding Domain (AR-LBD) | Inactive | 0.99 |
| Respiratory toxicity | Active | 0.57 | Aromatase | Inactive | 0.85 |
| Cardiotoxicity | Inactive | 0.82 | Estrogen Receptor Alpha (ER) | Inactive | 0.72 |
| Carcinogenicity | Active | 0.63 | Estrogen Receptor Ligand Binding Domain (ER-LBD) | Inactive | 0.99 |
| Immunotoxicity | Active | 0.69 | Peroxisome Proliferator Activated Receptor Gamma (PPAR-Gamma) | Inactive | 0.91 |
| Mutagenicity | Active | 0.53 | Heat shock factor response element (HSE) | Inactive | 0.93 |
| Cytotoxicity | Inactive | 0.72 | Mitochondrial Membrane Potential (MMP) | Inactive | 0.62 |
| Phosphoprotein (Tumor Supressor) p53 | Inactive | 0.77 | Glutamate N-methyl-D-aspartate receptor (NMDAR) | Inactive | 0.96 |
| ATPase family AAA domain-containing protein 5 (ATAD5) | Inactive | 0.62 | alpha-amino-3-hydroxy-5-methyl-4-isoxazolepropionate receptor (AMPAR) | Inactive | 0.98 |
| Transtyretrin (TTR) | Active | 0.5 | Kainate receptor (KAR) | Inactive | 0.99 |
| Ryanodine receptor (RYR) | Inactive | 0.84 | Achetylcholinesterase (AChE) | Inactive | 0.88 |
| GABA receptor (GABAR) | Inactive | 0.81 | Constitutive androstane receptor (CAR) | Inactive | 0.98 |

warranting further experimental validation. Overall, the balanced profile, combined with the absence of major safety alerts, supported AMG-900 as a promising scaffold for anticancer drug repurposing.

## 4. Discussion

Cancer occurs when cells grow out of control and spread, making it one of the world's leading causes of death [93]. Breast, ovarian, and colorectal cancers are among the most common and deadly cancers, each influenced by genetic changes and lifestyle or reproductive factors. Although these cancers are usually studied separately, they share important molecular pathways that may be targeted together. In this study, we identified common key genes across these three cancers using bioinformatics approaches. We then used computer-based drug repurposing to find existing drugs that could potentially act on multiple cancer targets, offering a faster and more cost-effective path toward improved cancer treatment.

Transcriptomics looks at all the RNA molecules made inside a cell, giving a clear picture of which genes are switched on and how strongly they are working [94]. In cancer research, this approach is especially useful because cancer cells behave differently from normal cells and show distinct gene expression patterns [95]. By comparing these changes, researchers can better identify key genes and pathways that may drive cancer development and progression. Although colorectal, breast, and ovarian cancers are usually studied as separate diseases, growing evidence suggests that they may share common molecular features [96]. To better understand these shared mechanisms, this study focused on identifying genes that show similar expression changes across all three cancer types and exploring whether they could be targeted using existing drugs. By analyzing transcriptomic data from independent GEO datasets (GSE45827 for breast cancer, GSE21510 for colorectal cancer, and GSE26712 for ovarian cancer), we observed a large number of significantly up- and downregulated genes in cancer tissues, as shown in the volcano plots (Fig 2A). By intersecting the results from all three datasets, we identified 128 common differentially expressed genes (cDEGs), visualized in the Venn diagram (Fig 2B). Together, the identification of these 128 shared genes suggests the presence of common molecular programs that

are consistently altered across breast, colorectal, and ovarian cancers, pointing to their potential involvement in fundamental cancer-related processes and supporting their relevance for downstream functional and therapeutic analyses. After identifying the genes that were commonly altered in all three cancers, we next explored how their protein products interact with each other. This step was important because proteins work as part of interconnected networks rather than in isolation [97]. The dense interaction pattern observed in the PPI network suggests that these proteins are functionally linked and take part in shared biological processes related to cancer development. By focusing on the most highly connected proteins within this network, we were able to highlight key molecules that may act as central regulators of multiple signaling pathways. Narrowing the analysis to these hub proteins helped reduce complexity and allowed us to concentrate on the most biologically relevant targets. Beyond protein interactions, we also examined how these hub proteins are regulated at both the transcriptional and post-transcriptional levels. Transcription factors control when and how strongly genes are expressed, while microRNAs fine-tune this process by suppressing or reducing protein production after transcription [98,99]. Identifying TFs such as FOXC1, GATA2, and E2F1, along with key microRNAs including hsa-miR-92a-3p, hsa-miR-124-3p, and hsa-miR-20a-5p, helped explain why certain hub proteins show consistent dysregulation across cancers. These regulators were highlighted because they showed strong network connectivity and influenced multiple hub genes, suggesting that they play central roles in coordinating shared molecular programs in breast, colorectal, and ovarian cancers. Building on these interaction and regulatory network findings, we further sought to understand the biological meaning of the shared hub genes by performing functional enrichment analysis. While protein–protein interaction and regulatory analyses reveal how genes and proteins are connected, enrichment analysis helps explain what these genes actually do inside the cell and which cellular systems are most affected. Therefore, Gene Ontology (GO) and KEGG pathway analyses were carried out to place the identified hub genes into a broader functional context. The enrichment analysis demonstrated that the shared hub genes are primarily involved in biological processes related to cell cycle regulation, mitotic progression, and energy-dependent cellular activities, all of which are fundamental to cancer development. In normal cells, these processes are tightly controlled to ensure accurate cell division. However, their enrichment across breast, ovarian, and colorectal cancers suggests that this regulatory balance is disrupted, allowing cancer cells to bypass normal checkpoints and proliferate uncontrollably. Such persistent activation of mitotic processes promotes rapid tumor growth and increases the likelihood of accumulating genetic errors [100]. Cellular component analysis further strengthened this connection by showing that many hub genes are localized to critical mitotic structures, including the centrosome, mitotic spindle, and nucleus. These structures are essential for proper chromosome alignment and segregation during cell division. Dysfunction within these components can lead to chromosome missegregation and aneuploidy, a hallmark of cancer that contributes to genomic instability and tumor heterogeneity [101]. The consistent enrichment of these components across all three cancer types indicates that mitotic structural defects represent a shared mechanism driving cancer progression. At the molecular function level, enrichment of ATP binding, kinase activity, and microtubule-associated functions highlights abnormal signaling and cytoskeletal regulation in cancer cells. Excessive kinase activity can override cell cycle checkpoints, while altered microtubule dynamics support continuous chromosome movement and division, collectively enabling sustained tumor cell proliferation [102]. In agreement with these findings, KEGG pathway analysis revealed significant enrichment of the cell cycle, oocyte meiosis, and p53 signaling pathways. Dysregulation of these pathways impairs DNA damage responses and tumor-suppressive mechanisms, ultimately promoting uncontrolled growth and cancer progression [103]. Overall, the integrated transcriptomic, proteomic, pan-cancer, and immune infiltration analyses consistently highlight AURKA, CCNB1, and CDK1 as key oncogenic regulators across multiple cancer types. Although AURKA, CCNB1, and CDK1 are not novel discoveries, they are well-established regulators of cell cycle progression and mitotic control in cancer biology, as identified through an extensive literature review [104–110]. Consistent with prior reports linking AURKA to mitotic spindle control, chromosomal instability, and poor prognosis, as well as active efforts to therapeutically target this kinase, our results corroborate its central role in tumor progression. Similarly, CCNB1 and CDK1 constitute a canonical mitotic complex whose dysregulation is associated with uncontrolled proliferation and

adverse clinical outcomes, and our data reinforce their importance within this disease context. Thus, the identification of these genes in our study reinforces the robustness of the analytical pipeline, as it successfully recapitulates known oncogenic drivers across BOC cancers. Importantly, their consistent dysregulation across multiple cancer types highlights their potential as common therapeutic targets and supports the concept of shared oncogenic mechanisms.

Their robust overexpression in tumor tissues, validated at both mRNA and protein levels, underscores their fundamental role in cell cycle dysregulation and tumor progression. The widespread pan-cancer significance further suggests that these hub genes act as common molecular drivers rather than cancer-type–specific markers. Importantly, the positive association between hub gene expression and CD8⁺T-cell infiltration indicates a potential link between proliferative signaling and immune-active tumor microenvironments. This dual involvement in tumor growth and immune contexture positions these targets as promising candidates not only for anticancer drug development but also for immuno-oncology–oriented therapeutic strategies. Based on the above-mentioned results and discussion AURKA, CDK1, and CCNB1 emerged as central regulators and selected for further investigation to develop multi-targeted drug candidates.

To explore this therapeutic potential, structure-based drug repurposing was performed to identify compounds capable of binding effectively to all three targets. Molecular docking analysis suggested that several repurposed drugs could interact with AURKA, CCNB1, and CDK1, highlighting their potential draggability. Among the screened compounds, AMG-900 exhibited strong binding affinity across all three proteins in docking analyses, forming multiple hydrogen bond interactions within their predicted active sites. The successful redocking of the co-crystallized ligand with low RMSD demonstrates that the chosen grid definition, protonation states, and docking parameters are appropriate for this binding site, thereby increasing confidence in the predicted poses and relative affinities of the screened compounds [71]. However, subsequent molecular dynamics (MD) simulations and MM-GBSA analysis revealed a more nuanced and target-specific behavior. While AMG 900 maintained stable binding with AURKA and CCNB1 over 500 ns simulations, supported by consistent RMSD, a compact radius of gyration, reduced residue level fluctuations (RMSF), and favorable binding free energies, the interaction with CDK1 appeared less stable. Specifically, CDK1 showed increased RMSD and RMSF upon ligand binding, indicating enhanced structural flexibility or potential destabilization. Moreover, the positive MM-GBSA binding free energy for the CDK1–AMG-900 complex suggests that the interaction is thermodynamically unfavorable under simulated conditions.

This discrepancy between docking and MD-based results indicates that the initially predicted high affinity for CDK1 may not translate into a stable complex in a dynamic, solvated environment. Instead, AMG-900 binding may induce conformational perturbations in CDK1 or represent a transient interaction rather than a stable inhibitory mode. In contrast, AURKA and CCNB1 demonstrated consistent structural stabilization, as further supported by PCA, DCCM, and free-energy landscape analyses, which indicated restricted large-scale motions and energetically favorable conformations.

The pharmacokinetics properties indicated acceptable drug-like properties with manageable safety concerns, although relatively low solubility may require formulation optimization. AMG-900 is not a novel compounds, it is a potent, orally bioavailable pan-Aurora kinase (A/B) inhibitor that interferes with mitotic progression by blocking histone H3 phosphorylation, ultimately leading to mitotic catastrophe and apoptosis in cancer cells [111]. Preclinical studies have shown that it can effectively suppress tumor cell proliferation across a broad range of cancer types, including drug-resistant and p53-dysregulated models, highlighting its strong potential for repurposing [112]. At the mechanistic level, AMG-900 induces cell cycle arrest, polyploidy, and chromosomal missegregation, leading to significant tumor growth inhibition in both *in vitro* and *in vivo* models [113]. Early clinical investigations further suggest that the drug has a manageable safety profile and favorable pharmacokinetics, with encouraging anti-tumor activity, particularly in chemotherapy-resistant cancers [114]. Moreover, its ability to act synergistically with microtubule-targeting agents makes it a promising candidate for combination-based therapeutic strategies, especially in aggressive cancers such as triple-negative breast cancer [115]. Although the toxicity profile of AMG-900 indicates moderate toxicity, including hepatotoxicity, neurotoxicity, and carcinogenicity, it suggests a non-benign toxicity profile that requires caution. Such adverse effects are commonly observed for

anticancer agents [116,117], where therapeutic efficacy against tumor cells outweighs manageable adverse effects; these should also be validated by experimental studies before recommending this compound as therapeutic for cancer treatment. Therefore, according to this study, AMG-900 should be considered a hypothesis-generating candidate, requiring rigorous experimental and clinical validation of both efficacy and safety before any therapeutic recommendation. Taken together, these findings suggest that AMG-900 could be a potential multi-target drug agent, particularly against AURKA and CCNB1, while its effect on CDK1 may be less direct or dependent on alternative mechanisms. Therefore, experimental validation, such as kinase inhibition assays, is necessary to confirm its functional activity against CDK1. This integrative computational approach underscores both the promise and limitations of in silico drug repurposing and highlights the importance of combining docking with dynamic and energetic analyses for more reliable target evaluation.

## 5. Limitations of this study

Though this research paper offers useful information on common oncogenic regulators in breast, ovarian, and colorectal cancers, it is fundamentally founded on computational studies, which are subjected to some limitations. The findings are based on publicly accessible datasets and in silico predictions, which might not be able to reflect the full biological complexity of tumor microenvironment and patient heterogeneity. In addition to this, although molecular docking, MM-GBSA, and molecular dynamics simulation provide more in-depth information on binding stability, but it cannot entirely substitute experimental validation like an enzyme inhibition assay, cell-based cytotoxicity test, or in vivo pharmacological test. The next step in the work should involve experimental and clinical validation of the identified targets and the repurposed drug AMG-900 with the aim of validating its effects of multi-target inhibition and its safety.

## 6. Conclusion

The current study demonstrates that breast, ovarian, and colorectal cancers share a set of core oncogenic regulators centered on cell cycle control through AURKA, CCNB1, and CDK1. These findings highlight those diverse malignancies may arise from common molecular dysregulations rather than isolated genetic events. Molecular docking identified AMG-900 as a compound with strong binding potential across all three targets. However, subsequent dynamic and energetic analyses revealed target-specific differences, with stable and favorable interactions observed for AURKA and CCNB1, while the CDK1 complex appeared comparatively unstable, supported by increased structural fluctuations and an unfavorable binding free energy. These results suggest a context-dependent interaction pattern of AMG-900 within this regulatory network, with potentially greater relevance for AURKA and CCNB1 compared to CDK1. While these findings provide supportive computational insights into multi-target interactions, they should be considered as an initial step toward understanding the therapeutic potential of AMG-900Accordingly, this study provides a foundation for experimental validation of the identified targets and interactions to confirm their functional and translational relevance across multiple cancer types. Overall, it highlights the potential of integrative hypothetical approaches in uncovering shared molecular targets for future therapeutic exploration.

## Supporting information

**S1 Fig. Sensitivity analysis of PPI network using high-confidence threshold.** (A) PPI network generated at a confidence score of 0.7, illustrating the refined interaction landscape after removal of low-confidence edges. (B) Top 10 hub genes ranked by degree centrality, highlighting the most connected and potentially significant nodes.
(DOCX)

**S2 Fig. Comparative expression profiling through Boxplot analysis of the cKHGs in various cancers and their pathological phases using TIMER2.0.** Red boxes indicate tumor tissues and blue boxes indicate normal tissues. Statistically significant differences are marked as *$P < 0.05$, **$P < 0.01$, and ***$P < 0.001$.
(DOCX)

**S3 Fig. Correlation of AURKA CCNB1 and CDK1 Expression with Immune Cell Infiltration Across CD8+T Cells.**
(DOCX)

**S4 Fig. Docking protocol validation by re-docking of native ligands (2JZ, LZ9, and 1QK) in (A) AURKA, (B) CCNB1, and (C) CDK1 showing RMSD values of 1.649, 2.858, and 0.825 Å, respectively.**
(DOCX)

**S1 Table. Target Protein Information's and Molecular Docking Parameters.**
(DOCX)

**S2 Table. List of Unique Differentially Expressed Genes (DEGs) from Three Datasets.**
(DOCX)

**S3 Table. Docking scores of all protein–drug complexes.**
(DOCX)

**S4 Table. Hydrogen bonding and hydrophobic interaction profiles of AMG-900 with AURKA, CCNB1, and CDK1 proteins obtained from molecular docking analysis.**
(DOCX)

## Acknowledgments

The authors would like to acknowledge editor and all reviewers for their valuable comments that help us to improve the quality of the manuscript.

## Author contributions

**Conceptualization:** Hriddhi Sarker, Md Ahad Ali.

**Data curation:** Hriddhi Sarker, Farhad Bin Farid, Marguba Kamrun, Esha Masud, Neladre Shaker Roy.

**Formal analysis:** Hriddhi Sarker, Farhad Bin Farid, Asif Ahmed, Mamun Miah, Neeraj Kumar.

**Investigation:** Hriddhi Sarker, Farhad Bin Farid, Esha Masud.

**Methodology:** Hriddhi Sarker, Md Ahad Ali.

**Project administration:** Md Ahad Ali.

**Resources:** Farhad Bin Farid, Marguba Kamrun, Mamun Miah, Neladre Shaker Roy, Neeraj Kumar.

**Software:** Hriddhi Sarker, Farhad Bin Farid, Marguba Kamrun, Mamun Miah, Neeraj Kumar.

**Supervision:** Md Ahad Ali.

**Validation:** Hriddhi Sarker, Asif Ahmed, Md Ahad Ali.

**Visualization:** Hriddhi Sarker, Marguba Kamrun, Esha Masud, Mamun Miah.

**Writing – original draft:** Hriddhi Sarker, Esha Masud, Asif Ahmed.

**Writing – review & editing:** Marguba Kamrun, Md Ahad Ali.

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
