## [Decision Letter · Decision Letter 0]

19 Mar 2026

PONE-D-26-01784Multi-omics and pan-cancer analysis revealed common molecular signatures to disclose multitargeted anticancer agents through network pharmacology approachPLOS One

Dear Dr. Ali,

Thank you for submitting your manuscript to PLOS ONE. After careful consideration, we feel that it has merit but does not fully meet PLOS ONE’s publication criteria as it currently stands. Therefore, we invite you to submit a revised version of the manuscript that addresses the points raised during the review process. Please submit your revised manuscript by May 03 2026 11:59PM. If you will need more time than this to complete your revisions, please reply to this message or contact the journal office at plosone@plos.org. Please include the following items when submitting your revised manuscript:

We look forward to receiving your revised manuscript.

Kind regards,

Chandrabose Selvaraj, Ph.D.

Academic Editor

PLOS One

Journal Requirements:

“This research has not received any funding”

Reviewers' comments:

Reviewer's Responses to Questions

**Comments to the Author**

1. Is the manuscript technically sound, and do the data support the conclusions?

Reviewer #1: Partly

Reviewer #2: Yes

Reviewer #3: Partly

2. Has the statistical analysis been performed appropriately and rigorously? 

Reviewer #1: N/A

Reviewer #2: Yes

Reviewer #3: Yes

3. Have the authors made all data underlying the findings in their manuscript fully available?

Reviewer #1: Yes

Reviewer #2: Yes

Reviewer #3: Yes

4. Is the manuscript presented in an intelligible fashion and written in standard English?

Reviewer #1: Yes

Reviewer #2: Yes

Reviewer #3: Yes

5. Review Comments to the Author

Reviewer #1: 1. The current docking study reports binding affinities for AMG-900 but these values are meaningless without a reference. Authors must include a known co-crystallized inhibitor or a clinically approved drug for each of the three targets (AURKA, CDK1, and CCNB1) to serve as a positive control. This is necessary to validate that the docking protocol can accurately reproduce known binding modes and that the new candidate truly shows superior or comparable affinity.

2. While the authors mention three independent repetitions in the methods, the results section must explicitly show the averaged data (mean ± SEM) for metrics like RMSD, RMSF, and SASA across these replicates. Providing the statistical variance between the three runs is critical for establishing the reliability of your stability claims.

3. The Authors should explicitly state the PDB IDs used for AURKA, CDK1, and CCNB1 and justify their selection.

4. A medium confidence score of 0.4 was used in the STRING database to construct the PPI network. In network pharmacology, a threshold of 0.4 can often introduce significant biological "noise." Authors should justify why this lower threshold was chosen instead of a high-confidence threshold (0.7), or perform a sensitivity analysis to see if the identified hub proteins (AURKA, CDK1, CCNB1) remain central at a higher confidence level.

5. The abstract labels AMG-900 as having "less-toxic properties". However, the predicted toxicity results in Table 5 show high probabilities for hepatotoxicity (0.6), neurotoxicity (0.7), and carcinogenicity (0.63).

6. Briefly justify the choice of the OPLS3e force field for this specific ligand-protein system

Reviewer #2: This is a well-structured and comprehensive manuscript describing a bioinformatics and drug repurposing study aimed at identifying common therapeutic targets and a multi-targeted drug candidate for breast, ovarian, and colorectal (BOC) cancers.

I am incorporating detailed analysis of the manuscript, including its strengths, weaknesses, and specific suggestions for improvement.

Overall Assessment

The manuscript presents a logical and rigorous computational workflow, from differential gene expression analysis to molecular dynamics simulations and ADMET prediction. The identification of AURKA, CCNB1, and CDK1 as common hub genes is well-supported, and the subsequent identification of AMG-900 as a potential multi-target inhibitor is compelling. The study is relevant to the fields of pan-cancer analysis, network pharmacology, and drug repurposing.

Major Strengths

1. Clear and Logical Workflow: The study design (Fig. 1) is excellent. It guides the reader through a step-by-step process from data mining to preclinical in-silico validation, making the research easy to follow.

2. Multi-Omics and Multi-Level Validation: The authors don't stop at identifying DEGs. They validate their key hub proteins (KHPs) using:

- PPI Networks: To find highly connected hub genes.

- Regulatory Network Analysis: To understand upstream control by TFs and miRNAs.

- Functional Enrichment: To confirm the biological relevance of the KHPs in cell cycle pathways.

- External Transcriptomic and Proteomic Validation: Using GEPIA2 (TCGA/GTEx) and UALCAN (CPTAC) to confirm overexpression at both the mRNA and protein levels in independent cohorts. This significantly strengthens the findings.

- Pan-Cancer Analysis: Using TIMER 2.0 to show the dysregulation of these genes across a wider range of cancers, reinforcing their broad oncogenic importance.

3. Rigorous Computational Chemistry: The drug repurposing and validation are thorough. The use of molecular docking, followed by extensive 500 ns molecular dynamics (MD) simulations, MM-GBSA binding free energy calculations, and PCA, provides a high level of confidence in the stability and strength of the proposed protein-ligand interactions.

4. Practical Relevance: The study concludes with a promising, well-vetted drug candidate (AMG-900) and a clear path forward for experimental validation, bridging the gap between computational prediction and laboratory testing.

Major Weaknesses and Suggestions for Improvement

1. Inconsistency in Drug Nomenclature

This is the most critical error that needs correction before publication. The abstract and introduction mention "AGM-900" (a typo), while the results, figures, and tables correctly refer to "AMG-900." This must be corrected throughout the document.** AMG-900 is a well-known pan-Aurora kinase inhibitor developed by Amgen. Using the correct name is essential for reader comprehension and literature searches.

- Action: Replace all instances of "AGM-900" with "AMG-900." Carefully check the abstract, introduction, and any other sections.

2. Discussion of CDK1 Results

The MD simulation and MM-GBSA results for CDK1 are ambiguous and require a more nuanced discussion.

- The Data:

- RMSD for CDK1 increased upon ligand binding (from 3.14 Å to 4.15 Å).

- RMSF for CDK1 increased upon ligand binding (from 1.314 Å to 2.072 Å).

- MM-GBSA binding free energy was positive (+3.03 kcal/mol).

- Current Interpretation: The manuscript states that the positive MM-GBSA value "may reflect transient or dynamic binding, possible induced-fit flexibility, or regulatory interactions, suggesting that AMG-900 could still influence CDK1 activity through non-stable but functionally relevant interactions." This interpretation is too optimistic and scientifically questionable. A positive binding free energy, by definition, suggests the interaction is thermodynamically unfavorable.

- Suggestion for Revision:

- Acknowledge the discrepancy more directly. The high binding affinity from docking (-9.4 kcal/mol) does not always translate to a stable complex in a dynamic, solvated environment.

- The increased RMSD and RMSF suggest that AMG-900 binding may induce conformational changes or destabilize the CDK1 structure, rather than stabilizing it.

- The positive MM-GBSA value strongly indicates that the complex is not stable under the simulation conditions.

- Revised Interpretation Example: "While molecular docking predicted a high affinity between CDK1 and AMG-900, subsequent MD simulations and MM-GBSA analysis revealed that this interaction may not be stable in a dynamic physiological environment. The increase in RMSD, higher residue fluctuations (RMSF), and a positive binding free energy suggest that AMG-900 binding leads to structural destabilization rather than forming a stable inhibitory complex. This could indicate that the docking pose was not representative of a true binding mode, or that the compound binds only transiently. Therefore, despite its strong effects on AURKA and CCNB1, the activity of AMG-900 against CDK1 may be less direct or require alternative binding mechanisms. This highlights the need for experimental validation, such as kinase inhibition assays, to confirm its functional effect on CDK1."

- This more critical interpretation strengthens the paper's scientific integrity and points to a specific area for future experimental work.

3. Immune Infiltration Analysis Rationale

The study focuses solely on CD8+ T-cells for immune infiltration analysis. While CD8+ T-cells are crucial, a brief justification for this choice is needed.

- Action: Add a sentence in Section 2.7.2 or the discussion explaining that CD8+ T-cells were chosen because they are the primary effectors of anti-tumor immunity, and their infiltration is a key prognostic marker and predictor of response to immunotherapies like immune checkpoint inhibitors. This context is already in the text but can be made more prominent.

4. Language and Grammar

The manuscript is understandable, but there are numerous grammatical errors and awkward phrasings that detract from its professional quality. A thorough editing pass is required.

- Line 15-16: "15 Cancer diseases are characterized by multifactorial disease..." -> "Cancer is characterized as a multifactorial disease..."

- Line 30: "...required additional experimental (in vivo and in vitro) and clinical 32 validations..." -> "...require additional experimental (in vivo and in vitro) and clinical validation..."

- Line 38: "Cancer is a heterogeneous group of diseases..." -> This is fine. But Line 39: "...develop destructively..." -> "...grow uncontrollably..." is better.

- Line 63-64: "lifestyle factors such as eating habits, lack of physical activity, smoking, and being 63 overweight can increase 64 the risk..." -> "...lifestyle factors such as diet, physical inactivity, smoking, and obesity can increase the risk..."

- Line 97: "We integrated the 95 method of drug repurposing..." -> "We integrated drug repurposing strategies..." (The line numbers are a bit messy in the provided text).

- Line 320: "...powerful impact on various 318 target proteins." -> "...significant impact on various target proteins."

- Line 410: "...using their corresponding accession code..." -> "...using their corresponding PDB IDs..."

- Line 566: "3.12.1 Physicochemical and ADME properties prediction" -> "3.12.1 Prediction of Physicochemical and ADME Properties"

- Line 584: The table refers to "CID 24856041" but the text and figures use AMG-900. Ensure consistency.

5. Figures and Tables

- Table 1: Good. Ensure the reference numbers `[40-43]` are correctly formatted in the final bibliography.

- Figure 1: Excellent.

-Figure 2: Good. The volcano plots are clear. The Venn diagram is essential.

- Figure 3: The PPI network is useful. Ensure the node labels for the top 10 hubs in 3B are legible in the final high-resolution version.

- Figure 5: The combination of GEPIA2 and UALCAN data is a strength. The figure legend should clearly state that (A-C) are mRNA data from GEPIA2 and (a-h) are protein data from UALCAN.

- Figure 6: The heatmap is a great way to visualize the top compounds. The title should be more descriptive, e.g., "Figure 6: Heatmap of binding affinity scores (kcal/mol) for the top 20 repurposed drugs against the target proteins AURKA, CCNB1, and CDK1."

- Figure 7: The 2D interaction diagrams from Discovery Studio are very informative. The legend should mention that the 3D structures are from PyMOL and the 2D interaction diagrams from BIOVIA Discovery Studio.

- Figure 8-11: These MD simulation figures are complex but well-presented. The captions are adequate.

- Table 4 & 5: Good.

6. Minor Point: Availability of Data

Consider adding a "Data Availability Statement" at the end of the manuscript, stating that all data used (GEO accession numbers, PDB IDs) are publicly available and that all results from the analysis are included in the manuscript and its supplementary files.

Summary of Key Revisions

1. Critical: Fix all instances of "AGM-900" to "AMG-900" throughout the entire document.

2. Major: Revise the discussion of the CDK1 MD and MM-GBSA results to be more scientifically critical and less speculative. Acknowledge the potential instability of this interaction.

3. Language: Perform a thorough copy-edit to correct grammatical errors and improve sentence flow.

4. Minor: Justify the focus on CD8+ T-cells for the immune infiltration analysis.

5. Consistency: Ensure "CID 24856041" (PubChem ID) is either replaced with "AMG-900" in the text (like Table 5) or its use is explained.

This manuscript presents a significant and well-executed piece of computational research. Addressing the points above, particularly the AMG-900 typo and the nuanced interpretation of the CDK1 data, will greatly enhance its clarity, scientific rigor, and chances of acceptance in a peer-reviewed journal.

Reviewer #3: Reviewer Comments to the Authors

The manuscript presents an interesting computational approach aiming to identify shared oncogenic targets across multiple cancer types and to propose a multi-target therapeutic strategy. The overall concept is appealing, and the integration of bioinformatics and molecular modeling is technically sound. However, several important issues need to be addressed before the work can be considered for publication.

1. Clarity, structure, and English language

Significant improvements are needed in terms of language clarity, structure, and overall readability.

The abstract requires substantial revision. For example, the opening sentence (“Cancer diseases are characterized by multifactorial diseases…”) is grammatically incorrect and unclear. In addition, it is not explicitly stated that AMG-900 is being proposed as a therapeutic candidate, which creates confusion about the study’s objective.

The introduction lacks proper structure and logical flow. It is not clearly divided into coherent sections, and several sentences appear disconnected from the main narrative. For instance, the reference to immunotherapy (around line 44) is not integrated into the rest of the discussion.

Importantly, the introduction does not adequately present the computational methods and rationale used in this study. Instead, it focuses broadly on cancer biology and biomarkers without clearly linking these elements to the methodology or objectives of the paper.

The figure legends should also be revised to improve clarity and ensure they are self-explanatory.

In contrast, the discussion section is generally well written and clear, which highlights that similar clarity should be achieved throughout the manuscript.

2. Conceptual positioning and novelty

The idea of identifying pan-cancer multi-targets and associated therapeutic candidates is interesting and relevant. However, the manuscript overstates its novelty.

The claim that such pan-cancer approaches are not commonly explored is incorrect. Similar strategies have been widely investigated, including studies focusing on key regulators such as TP53, MAPK pathways, and cell-cycle regulators.

The authors should revise these statements and better position their work within the existing literature.

3. Biological interpretation of targets

The three identified targets (AURKA, CCNB1, CDK1) are well-established oncogenic drivers, particularly in the cancer types studied.

The manuscript does not sufficiently acknowledge the extensive prior knowledge regarding these targets.

The discussion should be expanded to contextualize these findings within existing oncology literature, rather than presenting them as novel discoveries.

Additionally, the proposed link between these targets and immune-related mechanisms is weak, not well supported, and appears speculative. This aspect should either be significantly strengthened with evidence or toned down.

4. Interpretation of AMG-900

The discussion of AMG-900 is problematic and requires substantial revision.

AMG-900 is not a novel candidate; it is a known pan-Aurora kinase inhibitor that has already been evaluated in oncology clinical trials.

Therefore, describing its use as “repurposing” is misleading.

Moreover, the manuscript does not mention that AMG-900 did not progress beyond early clinical stages, partly due to toxicity concerns, which is highly relevant when proposing it as a therapeutic candidate.

These points must be clearly acknowledged, and claims regarding its therapeutic potential should be significantly tempered.

5. Strength of conclusions

While the computational approach is interesting, the results are not sufficient to support the strength of the claims made.

The conclusions rely entirely on in silico analyses without experimental validation.

Several statements (e.g., regarding therapeutic applicability and cross-cancer efficacy) are overstated and potentially misleading.

The authors should revise the manuscript to present their findings as hypothesis-generating rather than conclusive.

Overall assessment

In summary, the study is based on an interesting concept and uses appropriate computational tools. However, the current presentation overstates novelty and translational impact, lacks sufficient contextualization within existing literature, and contains issues in clarity and structure.

Substantial revision is required to:

- Improve language and organization

- Accurately position the work within the field

- Moderate claims regarding novelty and therapeutic relevance

- Provide a more balanced and evidence-based interpretation of the results

6. PLOS authors have the option to publish the peer review history of their article (what does this mean?). If published, this will include your full peer review and any attached files.

Reviewer #1: No

Reviewer #2: No

Reviewer #3: **Yes:** Coralie Ebert

---

## [Author Response · Author response to Decision Letter 1]

8 Apr 2026

Responses to the Reviewers

Reviewer #1:

Comment 1: The current docking study reports binding affinities for AMG-900 but these values are meaningless without a reference. Authors must include a known co-crystallized inhibitor or a clinically approved drug for each of the three targets (AURKA, CDK1, and CCNB1) to serve as a positive control. This is necessary to validate that the docking protocol can accurately reproduce known binding modes and that the new candidate truly shows superior or comparable affinity.

Response: We thank the reviewer for this important suggestion regarding the validation of docking protocol. In the revised manuscript, we have now included co-crystallized inhibitors, 2JZ (AURKA), LZ9 (CCNB1), and 1QK (CDK1), as positive controls and performed re-docking to validate the docking protocol. The obtained RMSD values confirm reliable pose prediction, and comparative docking shows that AMG-900 exhibits comparable or better binding affinity. These results have been added to the revised manuscript. Please see lines 506-517, 819-822, and S4 Fig for more details about the redocking and positive control docking values.

Comment 2: While the authors mention three independent repetitions in the methods, the results section must explicitly show the averaged data (mean ± SEM) for metrics like RMSD, RMSF, and SASA across these replicates. Providing the statistical variance between the three runs is critical for establishing the reliability of your stability claims.

Response: We sincerely appreciate the reviewer’s suggestion. We have now reported all the average data for MD metrics (such as, RMSD, RMSF, Rg, and SASA) as mean ± standard error of the mean (SEM) from three independent runs to ensure statistical reliability, and these have been included in the revised manuscript. Please see lines 544-551 and Table 3.

Comment 3: The Authors should explicitly state the PDB IDs used for AURKA, CDK1, and CCNB1 and justify their selection.

Response: We thank the reviewer for this insightful comment. We have now added an explanation in our revised manuscript about the PDB IDs (3HA6, 4Y72, and 6GU6) selection criteria for the target proteins AURKA, CCNB1, and CDK1, respectively. These structures were selected based on their preparation methods, high resolution, presence of co-crystallized ligands, absence of mutation, and missing residues, making them suitable for reliable docking and MD simulations. Please see lines 270-274 and 494-496, for clear justification regarding the selection of PDB structures.

Comment 4: A medium confidence score of 0.4 was used in the STRING database to construct the PPI network. In network pharmacology, a threshold of 0.4 can often introduce significant biological "noise." Authors should justify why this lower threshold was chosen instead of a high-confidence threshold (0.7), or perform a sensitivity analysis to see if the identified hub proteins (AURKA, CDK1, CCNB1) remain central at a higher confidence level.

Response: We thank the reviewer for this valuable suggestion. We initially used a medium confidence threshold (0.4) to capture a broader interaction network. Upon applying a higher confidence threshold (0.7), the PPI network became more stringent and fragmented, as weaker interactions were excluded. However, the key functional clusters, particularly those centered around CDK1–CCNB1 and MAD2L1–BUB1/BUB1B, remained clearly preserved. Importantly, the identified hub genes (CDK1, CCNB1, and AURKA) were still retained, although some appeared with fewer connections due to increased stringency. This indicates that our hub gene selection is robust and not driven by low-confidence interactions. The updated network has been included in the revised manuscript. Please see new added lines 382-384 and S1 Fig.

Comment 5: The abstract labels AMG-900 as having "less-toxic properties". However, the predicted toxicity results in Table 5 show high probabilities for hepatotoxicity (0.6), neurotoxicity (0.7), and carcinogenicity (0.63).

Response: We sincerely thank the reviewer for this observation. While AMG-900 shows some predicted toxicities (hepatotoxicity 0.6, neurotoxicity 0.7, and carcinogenicity 0.63), such moderate toxicities are commonly observed for anticancer agents, where therapeutic efficacy against tumor cells outweighs manageable adverse effects. Therefore, we have replaced “less-toxic properties” in the abstract with “manageable toxicity profile typical of anticancer agents” to more accurately reflect its expected safety context. Please see lines 34-35, and 847-859.

Comment 6: Briefly justify the choice of the OPLS3e force field for this specific ligand-protein system.

Response: We thank the reviewer for this valuable suggestion. The OPLS3e force field was selected for molecular dynamics simulations due to its high accuracy in modeling both protein and ligand systems, particularly in drug-like chemical space. OPLS3e has been extensively parameterized and validated for small molecule–protein interactions and is known to provide improved representation of torsional profiles, partial charges, and conformational energetics compared to earlier OPLS versions. Importantly, it is optimized for pharmacologically relevant compounds, making it highly suitable for evaluating protein–ligand stability and binding behavior in drug repurposing studies. Therefore, its use in this study ensures reliable and physiologically relevant simulation outcomes for the investigated ligand–protein complexes. We also added such statement in our revised methodology section, Please see lines 305 – 309.

Reviewer #2:

Comment 1: Inconsistency in Drug Nomenclature: This is the most critical error that needs correction before publication. The abstract and introduction mention "AGM-900" (a typo), while the results, figures, and tables correctly refer to "AMG-900." This must be corrected throughout the document. AMG-900 is a well-known pan-Aurora kinase inhibitor developed by Amgen. Using the correct name is essential for reader comprehension and literature searches. Action: Replace all instances of "AGM-900" with "AMG-900." Carefully check the abstract, introduction, and any other sections.

Response: We thank the reviewer for highlighting this important issue. We sincerely apologize for the inconsistency in drug nomenclature. All instances of “AGM 900” have now been corrected to “AMG 900” throughout the manuscript, including the Abstract and Introduction.

Comment 2: Discussion of CDK1 Results-

The MD simulation and MM-GBSA results for CDK1 are ambiguous and require a more nuanced discussion.

- RMSD for CDK1 increased upon ligand binding (from 3.14 Å to 4.15 Å).

- RMSF for CDK1 increased upon ligand binding (from 1.314 Å to 2.072 Å).

- MM-GBSA binding free energy was positive (+3.03 kcal/mol).

- Current Interpretation: The manuscript states that the positive MM-GBSA value "may reflect transient or dynamic binding, possible induced-fit flexibility, or regulatory interactions, suggesting that AMG-900 could still influence CDK1 activity through non-stable but functionally relevant interactions." This interpretation is too optimistic and scientifically questionable. A positive binding free energy, by definition, suggests the interaction is thermodynamically unfavorable.

- Suggestion for Revision:

Acknowledge the discrepancy more directly. The high binding affinity from docking (-9.4 kcal/mol) does not always translate to a stable complex in a dynamic, solvated environment. The increased RMSD and RMSF suggest that AMG-900 binding may induce conformational changes or destabilize the CDK1 structure, rather than stabilizing it. The positive MM-GBSA value strongly indicates that the complex is not stable under the simulation conditions. Revised Interpretation Example: "While molecular docking predicted a high affinity between CDK1 and AMG-900, subsequent MD simulations and MM-GBSA analysis revealed that this interaction may not be stable in a dynamic physiological environment. The increase in RMSD, higher residue fluctuations (RMSF), and a positive binding free energy suggest that AMG-900 binding leads to structural destabilization rather than forming a stable inhibitory complex. This could indicate that the docking pose was not representative of a true binding mode, or that the compound binds only transiently. Therefore, despite its strong effects on AURKA and CCNB1, the activity of AMG-900 against CDK1 may be less direct or require alternative binding mechanisms. This highlights the need for experimental validation, such as kinase inhibition assays, to confirm its functional effect on CDK1. This more critical interpretation strengthens the paper's scientific integrity and points to a specific area for future experimental work.

Response: We thank the reviewer for such insightful and constructive comment. We agree that the previous interpretation of the CDK1 results was overly optimistic. Accordingly, the discussion has been revised to more accurately reflect the thermodynamic and dynamic findings. Specifically, we now acknowledge that the positive MM-GBSA binding free energy indicates an unfavorable interaction, and that the increased RMSD and RMSF suggest structural destabilization upon ligand binding. The revised text clarifies that, despite the strong docking score, the CDK1–AMG 900 complex may not remain stable under dynamic conditions and could represent transient or non-representative binding. We have also emphasized the need for further experimental validation (e.g., kinase inhibition assays) to confirm any functional relevance. These revisions have been incorporated in the Abstract (lines 31–34), Discussion (lines 822–837), Result (lines 653-656), and Conclusion section to ensure consistency throughout the manuscript.

Comment 3: Immune Infiltration Analysis Rationale

The study focuses solely on CD8+ T-cells for immune infiltration analysis. While CD8+ T-cells are crucial, a brief justification for this choice is needed. Action: Add a sentence in Section 2.7.2 or the discussion explaining that CD8+ T-cells were chosen because they are the primary effectors of anti-tumor immunity, and their infiltration is a key prognostic marker and predictor of response to immunotherapies like immune checkpoint inhibitors. This context is already in the text but can be made more prominent.

Response: We thank the reviewer for this insightful comment. We agree that the rationale for focusing specifically on CD8⁺ T cells could be made more explicit. While this context was already present in the original text, we have revised the section 2.7.2 to improve clarity and emphasis. Specifically, we have added a sentence explicitly stating that CD8⁺ T cells are the primary effectors of anti-tumor immunity and serve as key prognostic biomarkers and predictors of response to immunotherapies, including immune checkpoint inhibitors. Please see the lines 262-264.

Comment 4. Language and Grammar

The manuscript is understandable, but there are numerous grammatical errors and awkward phrasings that detract from its professional quality. A thorough editing pass is required.

• Line 15-16: "15 Cancer diseases are characterized by multifactorial disease..." -> "Cancer is characterized as a multifactorial disease..."

• Line 30: "...required additional experimental (in vivo and in vitro) and clinical 32 validations..." -> "...require additional experimental (in vivo and in vitro) and clinical validation..."

• Line 38: "Cancer is a heterogeneous group of diseases..." -> This is fine. But Line 39: "...develop destructively..." -> "...grow uncontrollably..." is better.

• Line 63-64: "lifestyle factors such as eating habits, lack of physical activity, smoking, and being 63 overweight can increase 64 the risk..." -> "...lifestyle factors such as diet, physical inactivity, smoking, and obesity can increase the risk..."

• Line 97: "We integrated the 95 method of drug repurposing..." -> "We integrated drug repurposing strategies..." (The line numbers are a bit messy in the provided text).

• Line 320: "...powerful impact on various 318 target proteins." -> "...significant impact on various target proteins."

• Line 410: "...using their corresponding accession code..." -> "...using their corresponding PDB IDs..."

• Line 566: "3.12.1 Physicochemical and ADME properties prediction" -> "3.12.1 Prediction of Physicochemical and ADME Properties"

• Line 584: The table refers to "CID 24856041" but the text and figures use AMG-900. Ensure consistency.

Response: We thank the reviewer for these helpful suggestions. We have thoroughly and carefully revised the manuscript to correct grammatical and typographical errors, improve sentence structure, and enhance overall clarity. All the highlighted sentences have been corrected as suggested, and consistency issues (e.g., “CID 24856041” vs. “AMG-900”) have been resolved throughout the manuscript. Please let us know if the reviewer has any further concern regarding these.

Comment 5: Figures and Tables

• Table 1: Good. Ensure the reference numbers [40-43] are correctly formatted in the final bibliography.

• Figure 1: Excellent.

• Figure 2: Good. The volcano plots are clear. The Venn diagram is essential.

• Figure 3: The PPI network is useful. Ensure the node labels for the top 10 hubs in 3B are legible in the final high-resolution version.

• Figure 5: The combination of GEPIA2 and UALCAN data is a strength. The figure legend should clearly state that (A-C) are mRNA data from GEPIA2 and (a-h) are protein data from UALCAN.

• Figure 6: The heatmap is a great way to visualize the top compounds. The title should be more descriptive, e.g., "Figure 6: Heatmap of binding affinity scores (kcal/mol) for the top 20 repurposed drugs against the target proteins AURKA, CCNB1, and CDK1."

• Figure 7: The 2D interaction diagrams from Discovery Studio are very informative. The legend should mention that the 3D structures are from PyMOL and the 2D interaction diagrams from BIOVIA Discovery Studio.

• Figure 8-11: These MD simulation figures are complex but well-presented. The captions are adequate.

• Table 4 & 5: Good.

Response: We have carefully revised all figures and tables according to the reviewer’s comments. The legend of Figure 5 now explicitly states that panels (A–C) represent mRNA expression data from GEPIA2, while panels (a–h) correspond to protein expression data from UALCAN. The title of Figure 6 has been made more descriptive to clearly reflect the binding affinity heatmap. For Figure 7, we clarify that all interaction diagrams (both 2D and 3D representations presented in the figure) were generated using BIOVIA Discovery Studio. Although PyMOL was used during the analysis stage for detailed structural visualization as mentioned in the Methods section, the final figures were prepared using BIOVIA Discovery Studio. All other figures (Figures 1, 2,3, 8–11) and Tables 4 & 5 have been carefully checked and finalized accordingly.

Comment 6: Minor Point: Availability of Data

Consider adding a "Data Availability Statement" at the end of the manuscript, stating that all data used (GEO accession numbers, PDB IDs) are publicly available and that all results from the analysis are included in the manuscript and its supplementary files.

Response: We thank the reviewer for the suggestion. A Data Availability Statement has been updated accordingly, specifying that all GEO datasets (GSE45827, GSE21510, GSE26712) and PDB structures (3HA6, 4Y72, 6GU6) are publicly available, with all analysis results included in the manuscript and Supplementary Material.

Reviewer #3:

Comment 1: Clarity, structure, and English language: Significant improvements are needed in terms of language clarity, structure, and overall readability.

Response: We sincerely thank the reviewer for their valuable comment. We have thoroughly revised the manuscript to improve language clarity, structure, and overall readability, including careful editing for grammar, sentence flow, and coherence throughout the main text. Please see our revised highlighted manuscripts for clarification.

Comment 2: The abstract requires substantial revision. For example, the opening sentence (“Cancer diseases are characterized by multifactorial diseases…”) is grammatically incorrec

---

## [Decision Letter · Decision Letter 1]

17 May 2026

Multi-omics and pan-cancer analysis revealed common molecular signatures to disclose multitargeted anticancer agents through network pharmacology approach

PONE-D-26-01784R1

Dear Dr. Ali,

We’re pleased to inform you that your manuscript has been judged scientifically suitable for publication and will be formally accepted for publication once it meets all outstanding technical requirements.

Kind regards,

Chandrabose Selvaraj, Ph.D.

Academic Editor

PLOS One

Additional Editor Comments (optional):

Reviewers' comments:

Reviewer's Responses to Questions

**Comments to the Author**

1. If the authors have adequately addressed your comments raised in a previous round of review and you feel that this manuscript is now acceptable for publication, you may indicate that here to bypass the “Comments to the Author” section, enter your conflict of interest statement in the “Confidential to Editor” section, and submit your "Accept" recommendation.

Reviewer #2: All comments have been addressed

Reviewer #3: All comments have been addressed

2. Is the manuscript technically sound, and do the data support the conclusions?

Reviewer #2: Yes

Reviewer #3: Yes

3. Has the statistical analysis been performed appropriately and rigorously? 

Reviewer #2: Yes

Reviewer #3: Yes

4. Have the authors made all data underlying the findings in their manuscript fully available?

Reviewer #2: Yes

Reviewer #3: Yes

5. Is the manuscript presented in an intelligible fashion and written in standard English?

Reviewer #2: Yes

Reviewer #3: Yes

6. Review Comments to the Author

Reviewer #2: The authors accepted all the critiques, made the requested corrections (especially toning down the CDK1 claims and fixing the drug name), and provided a detailed point-by-point response.

Reviewer #3: (No Response)

7. PLOS authors have the option to publish the peer review history of their article (what does this mean?). If published, this will include your full peer review and any attached files.

Reviewer #2: No

Reviewer #3: **Yes:** Coralie Ebert

---

## [Editor Report · Acceptance letter]

PONE-D-26-01784R1

PLOS One

Dear Dr. Ali,

I'm pleased to inform you that your manuscript has been deemed suitable for publication in PLOS One. Congratulations! Your manuscript is now being handed over to our production team.

Kind regards,

on behalf of

Dr. Chandrabose Selvaraj

Academic Editor

PLOS One